# A qualitative study about how families coped with managing their well-being, children's physical activity and education during the COVID-19 school closures in England

Lisa Woodland[1,2]*, Ava Hodson[3], Rebecca K. Webster[4], Richard Amlôt[2,5], Louise E. Smith[1,2], G. James Rubin[1,2]

1 Department of Psychological Medicine, King's College London, London, United Kingdom, 2 NIHR Health Protection Research Unit in Emergency Preparedness and Response, England, 3 Department of War Studies, King's College London, London, United Kingdom, 4 Department of Psychology, University of Sheffield, Sheffield, United Kingdom, 5 Behavioural Science and Insights Unit, UK Health Security Agency, Salisbury, United Kingdom

* lisa.woodland@kcl.ac.uk

**Data Availability Statement:** Our data is deposited in the King's College London research data repository, KORDS, (https://doi.org/10.18742/

## Abstract

In 2020, schools in England closed for six months due to COVID-19, resulting in children being home-schooled. There is limited understanding about the impacts of this on children's mental and physical health and their education. Therefore, we explored how families coped with managing these issues during the school closures. We conducted 30 qualitative interviews with parents of children aged 18 years and under (who would usually be in school) between 16 and 21 April 2020. We identified three themes and eight sub-themes that impacted how families coped whilst schools were closed. We found that family dynamics, circumstances, and resources (Theme 1), changes in entertainment activities and physical movement (Theme 2) and worries about the COVID-19 pandemic (Theme 3) impacted how well families were able to cope. A key barrier to coping was struggles with home-schooling (e.g., lack of resources and support from the school). However, parents being more involved in their children's personal development and education were considered a benefit to home-schooling. Managing the lack of entertainment activities and in-person interactions, and additional health worries about loved ones catching COVID-19 were challenges for families. Parents reported adverse behaviour changes in their children, although overall, they reported they were coping well. However, pre-existing social and educational inequalities are at risk of exacerbation. Families with more resources (e.g., parental supervision, access to green space, technology to facilitate home-schooling and no special educational needs) were better able to cope when schools were closed. On balance, however, families appeared to be able to adapt to the schools being closed. We suggest that policy should focus on supporting families to mitigate the widening health and educational gap between families with more and less resources.

20243988). The data include sensitive information and cannot be made openly available. Requests for access to relevant excerpts of the transcripts will be considered if made by academic teams willing to sign a standard access agreement and should be addressed to the King's College London research data repository team at research.data@kcl.ac.uk.

**Funding:** This study was funded by the Economic and Social Research Council [grant number ES/P000703/1] and National Institute for Health and Care Research Health Protection Research Unit (NIHR HPRU) [grant number NIHR200890] in Emergency Preparedness and Response, a partnership between the UK Health Security Agency, King's College London and the University of East Anglia. The views expressed are those of the author(s) and not necessarily those of the NIHR, UKHSA or the Department of Health and Social Care. For the purpose of open access, the author has applied a Creative Commons Attribution (CC BY) licence to any Author Accepted Manuscript version arising. The funders had no role in study design, data collection, data analysis, data interpretation, or writing of the manuscript. The corresponding author had full access to all the data and had final responsibility for the decision to submit for publication.

**Competing interests:** GJR, RA and LS participated in the UK's Scientific Advisory Group for Emergencies, or its subgroups. These groups did not fund the study or authors. RA is an employee of the UK Health Security Agency. The author interests do not alter our adherence to PLOS ONE policies on sharing data and materials.

## Introduction

In England, schools and childcare facilities closed to most children on 23 March 2020 to reduce the spread of COVID-19 [1] with no indication of when they would re-open. However, schools were kept open for vulnerable children (e.g. children who had special educational needs, formally recognised in a "health care plan") and children of key workers (roles critical to the COVID-19 response) [2]. Further measures were also implemented to minimise peoples' interactions with others and to reduce COVID-19 transmission. Non-essential shops were closed, people were asked to work from home if they were able, and everyone was instructed not to socialise with anyone from outside their household [3,4]. The public were asked to leave home as infrequently as possible, only for essential items, and to keep two meters away from people from other households. People could leave home to exercise, although no more than once a day, and they had to stay in their local area. "Lockdown" was a term commonly used to describe these restrictions. "Furlough" (a system whereby the Government paid organisations 80% of their employees' salaries for hours not worked) was introduced to support those who were unable to work due to the restrictions [5,6].

Under these rules families in England entered a unique situation where they were required to stay within their homes, as much as possible and educate their children at home for an extended period of time. Fear, uncertainties and changes in routine as a result of COVID-19 are likely to have had adverse impacts on children's health [7]. To cope during this time, families would have had to adapt to the unique situation to be able to manage home-schooling and the lockdown guidance. Individuals who were unable to adapt to the change in situation risked distress. Research suggests that long-term distress affects relationships within the family, places strain on marital, parent-child and sibling relationships and can create a hostile home environment [8]. Such family conflict is also associated with mental health problems in children [9] that can follow into adulthood [10].

As well as potential mental health impacts, the pandemic resulted in the largest disruption to children's schooling since the second world war [11]. Schools provide children with education and access to health services (e.g., school nurses, specialist therapy services), a safe environment, nutritious food (children in low-income households can access free school meals), and interpersonal, social and occupational opportunities. As well as access to equipment, space and time for physical activity. Children are advised to engage in moderate to vigorous intensity physical activity for at least 60 minutes a day [12]. Physical activity is key to maintaining children's mental and physical health [13–15]. Loss of access to the benefits schooling brings resulted in widespread concern about the adverse mental and physical effects of the school closures on children [13,16]. The school closures are also likely to have reduced children's access to their support networks. Social support from peers and teachers increases children's school satisfaction, which is a key facilitator in children's cognitive and emotional development [17]. These risks to children's health and education are likely to have been increased for children who were already vulnerable. A UK study found that children from ethnic minority backgrounds, who had special educational needs and on free school meals were more likely to have emotional and peer relationship problems [18]. Children were also more at risk of mental health problems when their parent had a mental health problem, low social economic status, was single, or had low social support [9]. Low social economic status has also been associated with poorer education [19–21] and physical health outcomes [22–24]. For these reasons, campaigns to re-open schools were common across England, although they did not re-open until the start of a new school year, six months after they closed [25–27].

While schools were closed, home-schooling was primarily conducted via remote learning, which should have included a combination of recorded or live direct teaching time, time for

pupils to complete activities independently and physical activity [28]. The Government recommended that teaching time, at a minimum was: three hours for Key stage 1 (five to seven-year-olds), four hours for Key stage 2 (seven to eleven-year-olds) and five hours for Key stage 3 and 4 (eleven to sixteen-year-olds) [29]. This would mean that children would receive the same amount of teaching via remote learning in a day that they would have received if they were in school. Still, children needed the resources they would have had at school to continue their education at home, such as learning resources (e.g., textbooks), supervision, and guidance with their schoolwork [30]. For these minimum requirements to be met when their children were home-schooled during the school closures, parents had to take on a more prominent role in their children's education (including physical education) and fulfil some of the responsibilities of teachers (e.g., engaging children with schoolwork). By understanding how families coped during this time, such as the techniques they used to adapt and reduce the strain on the family system, the difficulties they faced, and how experiences differed due to family characteristics (e.g., age and number of children in the household), policymakers can identify what systems could be implemented in future to ease families' difficulties if schools need to be closed unexpectedly and for a long period of time. Therefore, in a future pandemic or severe disease outbreak when school unexpectedly close [31] authorities can better support families and mitigate the long-term health and educational impacts on children.

Research now suggests that the impact of the school closures and lockdown measures had profound impacts on children's mental and physical health and education and affected family life. However, little is known about families' lived experiences of coping with these challenges, particularly at the start of the pandemic, when uncertainty was high. In this study, we investigated how families managed i) family well-being, iii) children's physical activity, and iii) education during the school closures in England, and how these factors affected family coping at the start of the COVID-19 lockdown.

## Materials and methods

### Design

We conducted one-to-one qualitative interviews with parents of children aged 18 and under (school-aged children).

### Participants

To be eligible participants were required to be over 18 years of age, live in England, and have parental responsibility for at least one child (18 years and under) who was not attending childcare, pre-school, or school due to COVID-19.

We subcontracted participant recruitment to a specialist qualitative, market research service, Angelfish Fieldwork [32]. Advertisement started 7 April 2020 until 21 April 2020 and 539 potential participants applied to participate in the study and were screened for eligibility. Forty-seven participants were screened via telephone and if successful the participant was invited to interview the next day. Participants who met the screening criteria were prioritised for interview according to pre-determined demographic quotas. Quotas were based on gender, ethnicity, marital status, employment status, income, level of education, living region, keyworker status, the number of children in the household, and children's age to ensure a diverse sample. We interviewed 30 participants, a sample size determined using the framework proposed by Fugard and Potts [33] to provide a high likelihood of identifying the most prevalent themes and reach data saturation. Eligible participants were confirmed for interview in quick succession to mitigate the risk of Government guidance changing before we had reached data saturation. Two participants that were invited for interview were not interviewed because

written consent had not been provided or because they cancelled their appointment: they were replaced with other volunteers.

## Interview outline

We used a semi-structured interview guide to explore how families coped with the school closures. The interview guide was drafted by LW and based on concepts derived from existing literature about the topic of interest and the expertise from all co-authors. Four parents, known to the authors, who had children in school or childcare before the school closures reviewed our initial interview guide. We amended questions, clarifying those that appeared challenging to answer based on their feedback.

At the start of the interview, parents were reminded that they could withdraw at any time, encouraged to describe their individual experience and asked to share any information about the topic that they felt was relevant. Parents were asked a series of questions about daily life, such as to describe their children's hobbies and how the family was spending their time since schools had closed. Parents were also asked how much time they had spent engaged in physical activity and home-schooling. The interviewer followed up on parents' responses to these questions to gain further information about these topics. Parents were also asked if there was "any aspect of your children's life that may have benefitted [them] from the current situation" and what they had "found to be the most challenging about the school closures."

The full guide is included in the supplementary materials (S1 Text).

## Procedure

Parents received information sheets detailing the aims and objectives of the interviews. Two female researchers with qualitative interview experience (LW and AH) conducted the interviews via telephone. Interviews lasted a mean of 75 minutes (range: 36–98 minutes) and took place between 16 and 21 April 2020. All interviews were audio-recorded and transcribed verbatim by a transcription company. Parents were reimbursed for their time with a £40 e-gift card.

The research was approved by the Psychiatry, Nursing and Midwifery Research Ethics Sub-committee at King's College London (MRSP-19/20-18349). Participants provided written consent before the interview and verbal consent at the start of the interview. Consent was provided for nine statements, for example confirming that they had read and understood the information sheet, knew that they could withdraw from the study at any time, and agreeing to the interview being audio recorded and shared with an external transcription company.

## Reporting

We have reported data following the standards for reporting qualitative research: a synthesis of recommendations [34] and the consolidated criteria for reporting qualitative research (COREQ) [35].

As part of a wider study about families' experiences during the COVID-19 pandemic, parents were also asked questions about their understanding and management of COVID-19-like symptoms [36] and adherence to the COVID-19 guidance [37], and the findings from these responses are reported elsewhere.

## Analysis

Analysis was conducted using Nvivo version 12 software [38]. LW analysed the data using thematic analysis, the six-phase approach recommended by Braun and Clarke [39]. An inductive

approach was used from a positivist epistemological position. Once all the interviews were complete LW listened to all the audio recordings and checked them against the transcripts. All transcripts were read for a second time, and notes were taken about key ideas. Transcripts were read in full for the third time, and data were inductively grouped into initial topics drawn from the notetaking phase. These initial topic groups were not distinct, and data could be assigned to multiple groups. Data within the topic groups were coded into initial codes, which were reviewed and re-coded three times. During each stage of re-coding, data not relevant to the study aims were removed from the analysis, and with the aim that the data would be coded exclusively to one topic group. Themes and sub-themes were defined after the re-coding phase was complete. Feedback from peer review resulted in a change in how the themes were structured. Not all authors were involved in every discussion although at least one other author was involved in creating the initial topic groups and reviewed the coding at each stage of the process. We resolved disagreements through discussion until an agreement was reached between all authors.

## Results

### Participants

Thirty parents who were mostly female (67%, n = 20) with a mean age of 39 (range: 24 to 64 years) were included in the study. Most parents were married/cohabiting (70%, n = 21) and had two or more children (70%, n = 21). Further demographic information is presented in Table 1. All participants had at least one child who was not attending school or childcare because of the pandemic, although six were not in childcare or school before the closures and were the younger sibling of another relevant child (total children n = 70). The children's ages ranged from two weeks to 18 years, with a mean age of eight (mode: 10 years, n = 8).

We identified three themes and eight sub-themes relating to how families coped with the COVID-19 lockdown: 1) families' dynamics, circumstances, and resources; 1.1) home-schooling; 1.2) available resources and social support to home-school; 1.3) time spent on education; 1.4) parent's employment status and family characteristics; 2) changes in entertainment activities and physical movement; 2.1) reduced social interactions and choice of entertainment activities; 2.2) physical activity; 3) worries about the COVID-19 pandemic; 3.1) media and information; 3.2) worries about health.

### Theme 1: Families' dynamics, circumstances, and resources

A summary of the factors described in theme 1 is presented in Table 2.

**1.1 Home-schooling.** Parents reported numerous frustrations and difficulties with home-schooling. Parents commonly reported it was challenging to home-school whilst managing their other responsibilities. Parents were worried about times they had prioritised other essential tasks over home-schooling and how this would impact their children. In addition, parents who had to home-school several children with different abilities, needs and schoolwork, reported increased difficulties.

> *"The age differences between my four, I mean that it's really difficult to get . . . to be able to . . . it's like I need four of me, to teach them. Because it's really hard to keep them . . . it feels a bit like spinning plates. Like, you go to one, 'How're you doing with this?' The other one calls you, so you're back over there, but at the same time you're still trying to help that one, and it's just kind of 'Oh, dear Lord, this is too much,' <laughs> 'this is really hard. . .'" (P08)*

Parents felt they did not have the adequate skills to teach their children and commonly claimed *"I'm not a teacher,"* which they worried could be to the detriment of their children.

**Table 1. Parent (n = 30) and children (n = 70) demographic information.**

| Demographic Information | | Frequency (%) |
|---|---|---|
| **Gender of Participant** | Female | 20 (67%) |
| | Male | 10 (33%) |
| **Ethnicity of Participant** | White | 20 (67%) |
| | Black, African, Caribbean, or Black British | 5 (16.5%) |
| | Mixed or Multiple ethnic groups | 3 (10%) |
| | Asian or Asian British | 2 (6.5%) |
| **Marital Status of Participant** | Married / Cohabiting | 21 (70%) |
| | Single / Separated | 9 (30%) |
| **Work Status of Participant** | Full-time (Working over 30 hours a week) | 20 (67%) |
| | Part-time (working 8–29 hours a week) | 5 (16.5%) |
| | Home-maker | 3 (10%) |
| | Student | 1 (3%) |
| | Maternity Leave | 1 (3%) |
| **Income of Participant** | Under £30,000 | 12 (40%) |
| | £30,000 - £50,000 | 8 (27%) |
| | Over £50,000 | 10 (33%) |
| **Level of Education of Participant** | $\leq$ A-level or equivalent | 12 (40%) |
| | $\geq$ Degree or equivalent | 18 (60%) |
| **Living Region of Participant**[*] | Yorkshire and the Humber | 5 (16.5%) |
| | East of England | 4 (13%) |
| | Greater London | 4 (13%) |
| | South West | 4 (13%) |
| | West Midlands | 4 (13%) |
| | North West | 3 (10%) |
| | South East | 3 (10%) |
| | East Midlands | 2 6.5% |
| | North East | 1 (3%) |
| **Keyworker status of the Participant** | No | 25 (83.5%) |
| | Yes | 5 (16.5%) |
| **Number of children in the Household** | 1 | 9 (30%) |
| | 2 | 12 (40%) |
| | 3 | 3 (10%) |
| | 4 and over | 6 (20%) |
| **Age of children (years)**[*] | 0–4 | 23 (33%) |
| | 5–8 | 13 (18.5%) |
| | 9–12 | 17 (24%) |
| | 13–16 | 14 (20%) |
| | 17–18 | 3 (4%) |
| **Usual education setting of children** | No childcare | 6 (8.5%) |
| | Nursery | 6 (8.5%) |
| | Pre-school | 5 (7%) |
| | Primary | 33 (47%) |
| | Secondary | 16 (23%) |
| | Sixth form/College | 4 (6%) |

[*]Percentages do not total 100 due to rounding errors.

**Table 2. The factors that affected how families coped with the COVID-19 school closures and lockdown measures in relation to Theme 1: Families' dynamics, circumstances, and resources.**

**Theme 1: Families' dynamics, circumstances, and resources**

**Sub-theme 1.1: Home-schooling**
• Parents were "not teachers," and they felt unable to adequately teach their children.
• Parents had three key feelings of "mum or parent guilt:" i) children had not learned enough ii) what they had taught their children was not right, iii) they were a bad parent.
• Home-schooling was a constant "battle" between parents and children. To mitigate this, parents appeared to adopt a child-led approach to ensure their children did "some work."
• Parents appreciated the autonomy they had to be able to educate their children about topics they wanted when home-schooling.
• Parents appeared to place a particular value in teaching their children skills that made them "self-efficient," such as, cooking and managing money which were skills parents felt were not taught in schools.

**Sub-theme 1.2: Available resources and social support to home-school**
• Parents were "reliant on technology," and home-schooling was commonly impeded by having inadequate resources to home-school.
• Support with home-schooling from partners and older siblings facilitated home-schooling.
• Children needed support from their school teachers to mitigate gaps in their education due to topics parents were unable to teach.
• Children with special educational needs were particularly disadvantaged; they did not have access to the "tailor-made" resources that they had when they were in school.

**Sub-theme 1.3: Time spent on education**
• Parents preferred to home-school their children in the mornings and for no more than four hours a day.
• Parents struggled to keep their children engaged with schoolwork and used various methods such as incentives to motivate them to complete their schoolwork.
• Schools that monitored children's work facilitated their education. However, these systems also put pressure on parents and children to complete an amount of work that they felt was unattainable.
• Children who were "self-motivated" and enjoyed the subjects they were learning were home-schooling themselves and required less guidance from their parents and school.
• Children who were revising for exams appeared to be less motivated to continue their education than children in formal lessons.

**Sub-theme 1.4: Parents' employment status and family characteristics**
• Parents working from home found it "challenging" to work and care for their children.
• Parents working but not from home struggled with not being available to support their children during the lockdown.
• Organisations that "looked after" their employees were key to helping parents cope with the difficulty of working during lockdown.
• Financial or job insecurity was a common burden on parents.
• Parents with only children appeared worried about how their children were coping and developing without having any social interactions with other children.
• Parents reported the main benefit of lockdown was that siblings and families had "bonded."

*"I'll be honest, I've got a masters degree in business. But that's all I've got. . .So, what we're doing is writing names and stuff like that. I can do that bit. That bit's fine. Jigsaws and recognising letters and numbers is fine. I can do that bit. But then, when it comes to like, 'Oh H\*\*, can you help me with my science test,' 'Oh, bloody hell, I don't even think I learnt this bit!'" (P15)*

Parents had three key worries about home-schooling, which appeared to create feelings of guilt: i) children had not learned enough, ii) what they had taught their children was not right, and iii) they were a bad parent;
i) children had not learned enough:

*"I worry . . . I mean in some ways I'm quite lucky. S\*\*\*'s only five, so the amount of schooling he's gonna miss, he'll be able to pick up quite quickly, but you do think what are they missing, am I doing a good enough job? When he goes back to school, are they gonna think we haven't bothered to do anything? I suppose it's how are you gonna be judged by the job that you've done looking after them?" (P12)*

ii) what they had taught their children was not right:

*"I haven't got a clue what I'm doing and I wouldn't know where to look to get ideas of resources to . . . it has definitely . . . plays my mind and that's to be honest I'm thinking about [inaudible word] going forwards is, we might need to be making a bit more of a conscious effort in terms of that side of things. I know she's getting out of bed and getting her exercise every day and that's happening. But academically, I'm a little bit worried in terms of how we're doing with it. And I'm not sure if we're doing it right."* (P20)

iii) they were a bad parent:

*"'I'm a bad mum,' that's what I just kept saying to my friends, 'I'm a bad mum.' I don't want to do schoolwork with him, because I was waking up with this anxiety, because I know at some point he's gonna cry. . . So, I'm just . . . trying to do what I can, but I do not feel he's nowhere near as ready for high school. And that's my worry, 'cause he's due to start high school in September and my worry is that, 'My Mum has messed me up!' <Laughs>"* (P11)

Parents commonly reported home-schooling was the most challenging aspect of being in lockdown. Parents reported there were often frustrations on both sides, i.e., from parents and children, which arose when they were home-schooling, and that home-schooling regularly resulted in arguments. Some parents reported it was a "battle" each time they tried to initiate home-schooling; children felt they did not need to do any schoolwork because they were not in school, and parents were unable to *"try and get them to understand the importance"* of home-schooling. Parents were concerned about the negative impacts on their relationship with their children due to the "battle" they had experienced. To mitigate the adverse effects on the parent-child relationship, parents suggested that the arguments due to home-schooling were not "worth it" and would not "force" home-schooling.

*"I don't want to force her to do stuff at the expense of our relationship, whilst we're also all stuck together in . . . it's one thing to do that when she has time away from me and she's got other alternatives of people to go to. But, when I'm her only source of entertainment during the day and her only support network, I don't want her to be in a position where she doesn't want to come and talk to me about other stuff, because she's mad at me for making her do schoolwork."* (P24)

Parents commonly used a "child-led" approach when teaching to mitigate arguments occurring and to get their children to do "some work." For example, *"we're trying at the moment just to make it fun and to drip feed a bit here and there when he's interested. I guess being led by him and when he's ready,"* and *"the fact that she's doing something she wants to do at a time she wants to do it."* A similar child-led approach was observed when parents reported they had a "relaxed approach," not a "strict schedule," and were "flexible" with home-school-ing. In addition, to help families cope with home-schooling, parents were trying "not to pres-sure" themselves when children were not engaged in schoolwork and suggested children *"have enough to worry about."*

*"I think we've been quite relaxed with the whole teaching situation. I've more focussed on the things that they enjoy doing. I'm not a qualified teacher, hands up to every qualified teacher out there, I don't know how they deal with 30 children in a classroom at once, because trying to wrangle two of them <laughs> is not always easy! So I have much respect for all the*

*teachers out there because I think they do a fantastic job. But . . . there are days where we've done no learning at all because they've both woke up on the wrong side of bed and the moment you put a workbook in front of them their like, 'Erm . . . no.'" (P26)*

However, some parents felt important topics were absent from the curriculum and found the autonomy to be able to teach their children these topics beneficial.

*"So we're using this time as well, to learn a few lessons. I have also been using this time, to step away from things outside of curriculums. So, I've been downloading activity booklets from the internet on Black history, icons such as Harriet Tubman, Marcus Garvey. I have been speaking to him about his history and learning about the Maroons, which we are descendants of. So, I'm just trying to educate him with things that are missed in the curriculum, but I think are equally as important to him as him knowing who Oliver Cromwell is and <laughs> your Churchills." (P11)*

Similarly, parents reported the benefits of "non-syllabus" and "non-curriculum" learning, such as learning from the environment rather than textbooks.

*"School is very syllabus-driven, but we're trying to use things around us to learn, as well. For example. . . we've discovered that it [fox] has its den underneath their [neighbours] shed, and then probably three days ago, we've seen that it's got six cubs. So, we've got this sort of nature case study on our doorstep. So, we've set them off to learn about foxes. . ." (P27)*

In addition, parents reported the importance of teaching children "home-skills" (e.g., cooking and cleaning), and "life-skills" (e.g., managing money, "well-being and personal development"), skills that parents felt made children "self-sufficient" and that were not taught adequately in schools.

*"Your child needs to be at home, relaxed, be their self, confident in who they are, and their ability. They don't need to learn 100 different pages of science or maths. Just do simple life skills with them. Teach them how to do the bed, teach them how to clean the floor, teach them how to bake a cake, or something like that, that you as a parent you feel happy doing. That's the skills that they'll need and all the other skills, yes, the education, the curriculum requirements, they can do that at school." (P22)*

Furthermore, parents appeared to reflect on how much housework they did and that they may not have identified the need for their children to learn these "self-sufficient skills" and observed the benefits of them learning these skills without the school closures.

*"I think so. I think she's become more confident in the kitchen . . . d'you know what I'm going to make her sound so stupid, but she wanted to make Super Noodles the other day and she put the hot water in and I was like 'Did you turn the stove on? Is there actually gas underneath it?' you know. 'Yeah, so you need to turn the stove on.' 'Oh yeah! . . . cutting the vegetables after you've taken the plastic off, or just little things like that. It's just like 'Hold on a minute, you don't know how to do these simple things?' and they're just things that I think that I just took for granted that she would know how to do but why would she know? If she's never had to do it." (P01)*

**1.2 Available resources and social support to home-school.** Parents reported that inadequate resources made home-schooling difficult. In particular, parents felt "reliant on

technology" because children commonly needed regular access to a computer, which affected the amount they could home-school. Schoolwork was commonly accessed via school platforms and apps, and lessons were moved online. Furthermore, parents working from home also needed regular access to a computer for their work. Parents commonly reported that they had not worked from home before or sporadically, such as when their children were off school due to illness, so most parents reported they did not have an office at home. Some employers pre-empted the lockdown and supplied office equipment which helped reduce the technology burden on families. However, parents commonly reported financial concerns about buying new equipment or could not buy what was needed to adequately home-school. Regardless of families' financial capacity, unstable internet connections and school websites crashing due to high demand were common barriers to home-schooling.

*'I'm a bit annoyed,' because nobody checked to see who actually had any form of internet access at home and if they could access Google Classroom. Because I don't have . . . my iPad doesn't work with Google Classroom because it's that old. My school . . . my work laptop doesn't work with Google Classroom because it is blocked to Google Drive. So we're having to use the phone app." (P11)*

Teachers were trying to support families who had resource difficulties. However, most of the solutions resulted in other challenges.

*"And things like, I mean we don't have a printer, so everything is just a million printouts. And so, I emailed the school and said, 'Is there a way around this?' And they said, 'Just write the answers down'. Well, that's fine but a lot of H\*\*\*'s fractions this morning was shading in the shapes. So, I'm sitting there, every day I've literally just been writing the worksheets out, which is not ideal." (P19)*

Families that had the resources to home-school also used technology as a learning tool to enhance their children's education, such as searching for information or watching educational programmes (e.g., *"I'm trying to get him to watch YouTube videos on how do you work out the area of a triangle)."* In contrast, some parents would discourage their children from using technology to help them with their schoolwork, suggesting they could not access the internet in class or for exams.

Parents often reported that they had bought new equipment to reduce the difficulties they had experienced from sharing equipment within the home. In addition, parents reported issues with equipment to home-school was a common source of arguments between siblings.

*"In terms of the house, we have Wi-Fi, but in their bedrooms, certainly A\*\*\* and B\*\*\*, so my eldest and middle one, they're slightly further away from where the hub is. So, when it comes to working on the laptop in their rooms, the connection's very sketchy, so they have to then come into one of the communal spaces, to do those bits of the work. And then that leads to fights over whose turn it is on the laptop <laughs>." (P27)*

Arguments were also common when families did not have the space to work, such as separate rooms for siblings and parents to work without distractions from their children.

Social support appeared to facilitate home-schooling. Parents with partners and older children who supported them with home-schooling alleviated the pressure on parents to teach. Parents suggested older siblings were especially useful in aiding their younger siblings' education because they had recently learned most of the topics and were creative when they taught.

*"Yeah, and then the other thing I've been doing as well is getting the two older ones also to work with my four-year-old as well, because they so enjoy that, it's nice for them to have an older brother or sister working with them, and it adds a bit of variation from either mum or dad doing it. And sometimes I feel as well that they can put things across in quite a fun way that I might not necessarily be able to. So, yeah, kind of using them as teachers as well for each other." (P18)*

Resources and support from the school were also vital, such as providing schoolwork and information about when the schools would re-open. Due to schools suddenly closing, teachers had to rapidly adapt and disseminate their lessons for online learning, and some schools were better than others at doing this. Some parents felt they had no guidance on how to educate their children, such as the topics appropriate to their children's age and ability.

*". . . or a guideline of what we should be doing or stuff like that, d'you know what I mean? Or . . . but, there's really just . . . we're kind of just fumbling about in the dark . . ." (P13)*

Furthermore, children who had extra support in school because of their individual needs appeared particularly disadvantaged in the support they received from schools. The additional support ceased, and schoolwork was not "tailor-made" to children's unique needs.

*"Yes, he was going out, a lot of also his and where he's behind is to do with his speech, and he was going out twice a week with a teaching assistant, like a communications group, and she was doing work with him, with other children who were also behind on their speech. And obviously he's not getting any of that support, so yeah, obviously that is going to then make him behind, because unfortunately we can't access that with being at home." (P18)*

The school closures also highlighted that schools are not solely used for educational purposes but also as a source of psychological support. Parents appeared to appreciate when schools kept in regular contact with them and when their children could contact the school. For example, a parent suggested that if their child was *"upset"* and *"don't want to talk to us. . .they've got somebody, their teachers,"* which reassured parents. However, some parents felt there were *"no actual support put in place,"* had sent *"token email[s],"* and *"deep down. . .there's nothing you [school] can do if I said, 'Oh, it's a really bad day.'"*

**1.3 Time spent on education.** The amount of education that was carried out within the home ranged between families, but parents commonly reported fewer than four hours a day. The mornings appeared to be the preferred time families allocated to home-schooling. For example, *"if they can get two or three hours done in the morning, well then, I think that's quite good"* and *"focus for a couple of hours in the morning with learning is better than nothing."* For children who needed more care and attention from their parents, usually pre-school age, the distinction between teaching their children and supervising them appeared less clear. Therefore, parents found it challenging to report the amount of teaching done.

Most parents reported difficulties with keeping their child's concentration on educational activities for what they felt was an acceptable amount of time. Parents suggested that it was difficult to keep their children working for prolonged periods without the structure of the school day.

*"But as a primary aged child, their attention isn't quite there yet, so he did the piece of work, or answered some questions and then, 'All right, that's done, I've finished! But actually there was more for the task that he needed to do, but he wouldn't hear any more. So, things like*

*that, 'cause then it's very difficult to bring him back onto the task, because in his mind, he's now completed it. Whereas in a classroom environment, it's, 'Actually no, there's still another 20 minutes of this class going on, so you'll carry on and do this!' But without that structure of a school day in terms of almost like the end of the lesson and the bell goes.'" (p27)*

Parents used incentives to encourage their children to complete their schoolwork, such as parents would reward their children once they had spent a specific amount of time on their schoolwork or had completed a certain amount. Children were commonly rewarded with an allowance of time that they could spend on non-educational activities (e.g., they could watch a film or play computer games).

In addition, parents in employment (excluding furlough) felt they could not monitor their child's schooling and spend time helping them with their schoolwork compared to if they did not have to work, which concerned parents.

*". . .there's an immense amount of guilt involved in us still, or me specifically working fulltime and trying to look after him, and pay him enough attention and there's also, you get a lot of . . . on social feeds or on newspapers and things, of amazing things you can do in the lockdown, or don't waste your time and do this and that . . . and there's a frustration that if you are still working and you haven't been furloughed or you haven't been . . . means that you can't, that you're missing out on some of the opportunities that other people are getting from it." (P12)*

The schools that monitored children's schoolwork were reported as a benefit because they reassured parents that there was some oversight in their child's education and motivated the child.

"not threat, but we always used to have to do his homework in the evening because his teacher would mark it the next day, so there was a reason why it had to be done. Whereas mummy says so doesn't always swing it with him."

Furthermore, children who could contact their teachers appeared more engaged with home-schooling, and a lack of support from their teachers was a barrier to home-schooling. The feedback from teachers' comments and from schoolwork that had been marked was vital in facilitating children's education.

*"But I suppose, a little bit more involvement from the marking side of things, would probably help the parents, I think. Even if it's sort of to then, start the week with, 'OK, all the work you did last week has come back from the teachers and this is some of the stuff they've said,' I suppose it helps to frame the following week's work, or the work that they're doing. 'cause otherwise it's just a bit of a black hole for the children. They do it and there's never any feedback to that, other than what we're doing with them. But again, we don't know if what we're doing is right, so we just go, 'Yeah, that looks good!'" (P27)*

When children were unable to contact their teachers, they left items blank or unable to progress when they did not "understand" their work. Without any resolution to these issues, children became demoralised, and these feelings increased as the number of outstanding issues increased. Parents who were able to provide feedback on their children's schoolwork somewhat mitigated the lack of support from teachers.

However, schools with systems that monitored children's work and progress also had some drawbacks. Some school monitoring systems were private, i.e., only a student and their parents

and teachers could view the pupil's schoolwork. But some systems were public, and everyone in the school or school year could view everyone else's work, which provided mixed reviews from parents. These open monitoring systems appeared to encourage children to complete their schoolwork and provide a support network that families found beneficial.

> *"And they also had to sort of like show you how to make a wand and all this sort of stuff. So that was quite nice. They both did that. And then what you do is you scan it and you upload it to the website, and the teachers . . . oh, you email it to them and then the teachers put it up on the website, so that all the work is seen. So they can see . . . the most helpful thing I think, is that they can see that other children are being made to do it as well <laughs>." (P08)*

However, the openness emphasised the amount of schoolwork or activities that other families had done for some parents, which made them feel inadequate. Similarly, schools that suggested numerous tasks and activities (publicly or privately) also had mixed reviews from parents. Some parents found this beneficial and suggested that the numerous tasks helped keep children entertained whilst continuing their education. However, some parents felt pressured to complete all the activities listed, resulting in parents and children worrying about any incomplete tasks.

> *"So for example the document that we have access to on a weekly basis is basically a gridded A4 sheet that gives you suggested activities from Monday to Friday and Monday might say mathematics, cover this, Tuesdays might say outdoor learning, Wednesday might be reading, writing but there'll be some various suggested things within that. . . They're all great suggestions and we could come up with hundreds ourselves, but it's the way that they're almost described, you inevitably feel pressured to cover them because you want to do the best for your child. . . you're automatically putting yourself into a box and setting yourself up to fail because there aren't enough hours in the day, even though we're at home for 90% of the day" (P25)*

Feeling supported and not pressured by their school mitigated some of the difficulties families had with home-schooling; school monitoring systems that provided entertainment, educational activities, and feedback on schoolwork helped keep children engaged.

Children's personal characteristics were also an essential factor that impacted the amount of home-schooling, irrespective of parent and teacher input. Some parents reported their children were "self-motivated," and parents were less concerned about being unable to monitor or assist them with their schoolwork. Alternatively, some children needed more "push" and direction in their schoolwork. In addition, the amount of schoolwork that schools expected differed, which parents found difficult to manage between siblings; children appeared to feel resentful when they had more schoolwork than their siblings.

> *"Do you know what I mean? It's like, 'Hang on a minute, how come I've got 15 pieces of homework today and C\*\*\*'s lying in bed playing Animal Crossing?'" (P06)*

Furthermore, the subject matter and children's interest in a subject appeared to influence children's motivation. Children interested in a topic would stay focused and engaged for extended periods than topics of less interest. In these instances, parents felt reassured that they could leave their children without monitoring their schoolwork. However, some parents were concerned that their children spent less time or ignored subjects they found less interesting and therefore, parents still had to motivate their children to engage with these topics.

Keeping motivated appeared particularly difficult for children who were revising for exams than children in 'standard' lessons or education. Families struggled to navigate and keep

children focused on revision when there were uncertainties about exams being held. Parents were uncertain if exams would be cancelled and, if so, how children's schoolwork would be graded. Children felt they were revising unnecessarily and were frustrated by the "waste" of time they had spent on revision but that they had to continue to revise, which left children feeling "deflated."

*"my son, who's 15, was due to sit his GCSEs this summer, so that's been a little bit of a bomb-shell, to be honest, for him. Yeah, so I was particularly interested in contributing from that perspective as well, because it's really, really had a massive impact. It's had a massive impact on all of us but I think for him particularly, he feels like the rug's been pulled from under his feet." (P06)*

Schools appeared to take different approaches to manage the uncertainty surrounding exams. Some schools informed children of their "final grade" if teachers graded their work. In these instances, some children were pleased with their expected grades, and parents suggested this reduced the amount they revised. These parents worried their children would be less prepared if the reported grades were not "final" because the grading system changed, or exams were to happen as usual.

*"Yeah, I'm not too worried about it. The only thing is, if they back-pedal and then they go, 'Oh, actually, this is gonna lift in three weeks and actually, you could sit your exams.' And I don't know if that's a potenti . . . I don't know how likely that is, but because it's not crystal clear, 'Here's your grades,' 'Here's your certificates,' I'm a bit worried." (P15)*

Alternatively, some schools confirmed that exams would not happen and children's grades, which helped families manage the uncertainty about exams. However, this was difficult for children who felt they could have done better in exams and did not have the opportunity to show their full potential.

*". . .P\*\*\* is more happy because he's been told that he'll get his literature result will get based on his mock and predicted grades, so he's expected more than he thinks that he would have done in the exam. R\*\*\* is a bit upset because, being profoundly deaf and that, obviously having learning disabilities, she thinks she could have done better in the exams than her mock exam, so . . . mixed feelings basically." (P14)*

**1.4 Parents' employment status and family characteristics.** Parents in employment appeared to have different stressors than parents who were unemployed or on furlough.

*". . . I just feel that when you're trying to do work or you're trying to do things that it's a little bit more harder, and I seem to have less patience because I'm trying to do stuff, than when the days where I'm not. I'm just like, 'You know what? I'm just leaving everything today and I'm just gonna play,' they're the best, least stressful days." (P02)*

Parents reported it was "challenging" and "really stressful" to concentrate on their work and look after their children when they were working from home. To combat this, parents commonly adjusted their working hours to work around their parental responsibilities. Parents reported they had formally changed their contractual hours, i.e., a change in work hours and pattern that parent and employer agreed upon. However, parents also changed their

work hours without this formal agreement. Instead, parents worked informal hours when they had *"some time to concentrate,"* such as when their children were asleep or engaged in an activity that did not require (or required minimal) parental supervision. As a result, parents often worked in the evenings, but they found it hard to concentrate because they were tired, such as, *"after a day with a child, my brain's fried."* The amount of parental supervision that children needed appeared to impact the amount parents changed their work hours; some parents reported they could mostly work uninterrupted. Therefore, they did not alter their work hours. Still, these parents had to stop working several times in a day when their children were "arguing," "hungry," or wanted help with something which demanded their parent's attention. Parents reported they had to constantly switch between being a parent to an employee, parent to a teacher, and teacher to an employee throughout the day, which was tiring, and tasks took longer than usual.

> *"But obviously because I'm doing my work and I'm running reports and working with high level of data, if he's stuck I have to break off to then help him. So then it's hard for me to get back into mine. So, my work kind of took a bit of a delay as well." (P11)*

Parents who were employed but did not work from home also changed their work hours, although this was commonly due to ensuring children had appropriate supervision at home whilst they were working. In these instances, parents commonly felt less able to support their children during the school closures, which they found difficult.

> *"I couldn't continue to work in the evenings, it's not something that suits me as a person, it doesn't suit my lifestyle as a person. I miss putting my children to bed at night. That is the biggest thing I've missed is the Monday to Friday, while I'm at work, I miss putting my kids to bed. But it's a small sacrifice on a temporary basis to be able to keep them safe and also do my job, which is important because my job keeps other people safe." (P26)*

Parents' employers impacted how parents coped with working and caring for their children. Parents who felt "looked after" and supported by their employers eased parents' burden, such as, *"[working] was giving me a bit of anxiety but work were quite supportive."* Organisations that quickly implemented procedures that adhered to the lockdown guidance (e.g., moving employees onto furlough and working from home) helped parents cope with the lockdown. Alternatively, organisations that were not *"forthcoming with the Government advice"* was a source of stress, especially for parents who had to "unnecessarily" work in the office and reported feeling "scared," "annoyed," and "anxious." Furthermore, how the government advice was implemented within organisations had caused "contention" between employees with children and those without. Parents experienced anger from non-parents who felt unfairly treated by organisations that prioritised parents in certain situations, such as organisations that prioritised furlough for employees with children.

The pressure on parents who were employed during the school closures was highlighted by parents who moved from working to furlough and parents who had taken annual leave. These parents reported "relief" and felt the *"psychological pressure has been lifted"* when they no longer had to work.

> *"And in the last week I took a few days off work, and I was kind of in two minds about it, [interviewers name], because you're like, I really don't want to use my annual leave to just sit in the living room, but I'll be honest in that it's been really stressful trying to be productive at work, keep an eye on all the kids' schoolwork and trying to keep them in a routine and make*

*sure everyone gets fresh air and make sure that we can find toilet rolls and you know, I've kind of just got to a point where I thought, 'This is just actually too much'. For the past week I've had some time off and it's been lovely." (P06)*

However, parents who were unemployed or had employment insecurity had stressors. Parents commonly reported the household income had decreased and struggled to make up for the loss in income, which was a source of worry.

*"My wife's a teacher and this year she's been doing supply teaching, which has been fantastic and a really good opportunity for her, but at the same time, as soon as they closed the schools, J\*\*\* didn't have any work. Because unlike other teachers, who will be going in and however they're balancing the workload with the keyworkers' children. But as a supply teacher, they're covering teachers that are off sick. And also, this year, I've been taking a bit of a career-break, so I haven't been working. Suddenly we were faced with zero income." (P27)*

Some parents reported financial difficulty because of nursery fees; some had to continue to pay the fees (and be in credit) otherwise their child would lose their nursery place when nurseries re-opened. Some parents reported that the nursery would not stop their payments or refund unused hours. However, not all parents reported financial worries, and some parents felt financially better off than before the school closures due to fewer expenses.

Parents reported that they had struggled to buy certain foods or had bought more expensive items because their usual cheaper brands were unavailable. Although this caused some parents "anxiety" and "worry," they reported they could adjust their shopping with alternative items. However, parents with children who needed nappies, baby food, and milk formula reported increased worry because they could not find suitable alternatives to these items.

Parents with one child seemed to worry about their child's development, loneliness, and boredom. In contrast, parents with multiple children appeared less concerned with these issues but commonly reported difficulties managing arguments between siblings. However, parents reported the "forced" interaction between siblings had some benefits and suggested their children had "got on better," building "stronger relationships," "friendlier towards each other," and had "bonded."

*"Yes, I guess . . . them sort of dedicating time to play with each other. I think that's a nice thing. They wouldn't normally do that, if they knew they only had a certain amount of time. They'd probably pick playing on the PlayStation or on the iPad over playing with each other. But now they're finally getting a bit bored with those things. So it's like 'Oh, let's actually play a real game, where we use our imaginations and play as friends.' So I'd like that to continue. I don't know if it will once they actually start seeing their actual friends again. You know, I mean . . . they have nice relationships as siblings, but it's nice seeing them actually make proper friendships with each other as well. So that's a nice thing, yeah." (P08)*

Similarly, large families or families with limited space reported *"being underneath each other all day 24/7 is not ideal"* and arguing more because they were *"in each other's space a lot."*
In addition, parents struggled without the "break" and "lack of rest" because of caring for their "children 24/7" and suggested it was "exhausting."

*"Yeah, the nine-year-old and the 10-year-old have definitely had some real hormonal meltdowns about different things. I think as well, because you're all clubbed together, normally you have a bit of a break, I feel myself as an adult, I could just do with having a break for five*

*minutes in a silent room, and I think that they feel the same. Normally you're going to school or you're going to work and have a bit of a break from each other...*" (P18)

Individual roles and responsibilities appeared to change within families to adapt to the new ways of working together, such as *"I used to come home around six o'clock and obviously the kids at school, now basically she's [partner] literally divided all the [house] work."* Similarly, 'traditional' gendered roles changed, *"I feel like those roles have probably levelled in terms of percentage of time and percentage of overall responsibility"* and *"I didn't realise how much me wife did, before she started this four 'til eight shift, with the kids and cooking tea and everything else"*

Furthermore, parents felt strained from *"putting on a brave face"* and trying to hide their worries from their children, and trying to keep "upbeat" and "positive" for them.

*"<Pause> I ... I'm trying. I had a bad day last weekend and I tried to not let C\*\*\* see it. And you know, I had a little bit of a cry and he says, 'Mum, are you OK?' And I says, 'Oh, my hay fever, it's getting in the way.' And he was happy with that. But the fact is, I was so low. I was ... it hit me and I could not stop crying. I couldn't get out of bed, I had to force myself to feed him. Then I burnt the dinner and I thought, 'Aaargh!' You know, so everything was just bad."* (P11)

## Theme 2: Changes in entertainment activities and physical movement

A summary of the factors described in theme 2 is present in Table 3.

**Table 3. The factors that affected how families coped with the COVID-19 school closures and lockdown measures in relation to Theme 2: Changes in entertainment activities and physical movement.**

| Theme 2: Changes in entertainment activities and physical movement |
| --- |
| **Sub-theme 2.1: Reduced social interactions and choice of entertainment activities** |
| • Parents commonly felt their children were coping well with lockdown and "adapting to the new situation." |
| • Some positive behaviour changes were identified, such as children were "less tired" and had "less anxiety." |
| • Due to school clubs and activities being restricted, parents felt there was more time during the day, and they were less "rushed" in the evenings, which resulted in a "relaxed" home environment. |
| • Families were able to spend more time together, which led to "understanding" each other better. |
| • Still, there were a range of adverse behaviour changes, such as children being "more anxious," "clingy" and "lashing-out." |
| • Families struggled with the closures of non-essential shops and entertainment organisations and being limited to 'in-house' activities. |
| • Struggles with coping with the lockdown increased when special events were cancelled, and life milestones were missed. |
| • Parents struggled to keep their children entertained due to limitations on the activities they could engage in, including socialising with friends and non-household family members. |
| • Socialising in person, via technology, and continuing online activities helped families cope. |
| **Sub-theme 2.2: Physical activity** |
| • Parents encouraged their children to exercise, suggesting it helped their children to "burn off energy" and maintain good moods and behaviours. |
| • Some children were exercising more than they would if they were in school and others less, which concerned parents. |
| • Parents would try to keep their children active by inventing new ways to keep them engaged with activities that result in physical movement. |
| • Physical movement was facilitated by technology, such as after school clubs and exercise classes that had moved online. |
| • Parents struggled to motivate themselves to stay active but were motivated to do so to encourage their children to keep active and maintain their family's well-being. |

## 2.1 Reduced social interactions and choice of entertainment activities

Most parents were surprised about how well their children were coping with adhering to the lockdown guidance, and suggested their children were "resilient," "accepting," *"taking it in their stride,"* and *"adapting to the new situation."* Parents also reported some positive behaviour changes that they had observed in their children, such as they were "less tired," "more interested in other things," "more relaxed," had "less anxiety," and *"more settled and sleeping better."*

> *"I really felt like I was gonna struggle with his behaviour and it was gonna be really hard work but I feel like he's kind of, he's been great with it. His behaviour's so much better, I don't know if it's because he's getting, he's got the whole attention of both of us and he's with us and he just feels more secure, attachment is securer. But he's been . . . obviously the odd time where we have to talk to him, but he has, he's been really good." (P02)*

Furthermore, parents commonly reported that the days were less "rushed" and "busy" than before the school closures. Families commonly reported that a week-day evening included lots of essential and mundane tasks, and then it was bedtime, such as helping with their children's homework, travelling home from work, picking up and or dropping off children at an after-school club or activity, making and eating dinner, and tidying up the dinner plates and house. Whereas parents felt that as well as spending more time together as a family which they enjoyed, their children were also more "chilled" and "relaxed" and without "school pressures," the atmosphere in the home was "nicer." In addition, parents reported changes that they had made due to the guidance that parents wanted to continue, such as for children to help them with the housework, taking regular family walks, *"spending the time to answer his questions instead of dismissing it,"* playing games as a family, being less wasteful with food, *"less of my being on my phone. . . and more time spent with them,"* and "more hygienic."

An apparent positive to the lockdown was that families could spend "more time together" and "quality time," which helped families cope with the lockdown. Families were having "diverse conversations," "sitting down together," "connecting as a family," watching TV and playing together more than they usually would. This time together resulted in parents perceiving their family to "understand" each other better.

> *"Yeah, funny, I was talking to . . . my mum was watching us, we were talking to her via this portal thing the other day, and she made the comment, and she just sort of said it off the cuff, but it's actually really stuck with me. She said, 'You've really got to know each other!' And I just thought that was really cute! <Laughs> And I kind of understand what she means. I feel like I've learnt more about him by being with him more." (P07)*

However, parents reported staying at home was difficult and "tricky" for families and suggested they were going "stir crazy." In addition, parents reported a range of adverse behaviour changes in their children: *"she's starting to get a little bit loud, a little bit silly;"* change in the amount of sleep (increase and decreased in the length of time, and increase in interrupted sleep); *"just fed up and she doesn't seem to be playing as much;"* "moaning;" "frustrated;" "become a lot quieter;" "short-tempered;" "more anxious;" "more babyish;" "can't be bothered" attitude; more clingy; more angry; an "attitude" (e.g., answering back); "more hyper;" "lashing out;" "more tantrums;" "stroppy;" "more irritable;" "not listening;" "more grumpy;" *and "more tears and upset over minor things."* Furthermore, it was particularly "confusing" for children who did not understand why they were unable to do the things that they usually

could. Some children thought their parents were punishing them because they had been "bad," rather than they could not do certain activities because of the guidance, which was "challenging."

Parents commonly reported that their family missed school, clubs, other non-essential places that were closed (e.g., museums, libraries, and theme parks) and the freedom to *"pop to the shops"* when they wanted. In particular, families felt "gutted" when special events that had been planned before the lockdown were cancelled, such as family holidays, birthday parties and bar mitzvahs.

> *"I booked seven months ago, back in October. Now I couldn't go basically. So it's affected obviously everyone, not just socially but mentally, physically and emotionally. . . Because my kids were looking forward to this holiday, since October". (P14)*

Furthermore, children in their final school year had the added difficulty of feeling they had missed key milestones.

> *"she had a summer play that she was going to be doing, and bowling, other fun activities that they had planned after SATs, and none of that now is going to be taking place. And obviously there's friends at the primary school that aren't going to be going to the same secondary school as her. . . I'm just trying to calm her down really as much as possible. I do hope, for her and for the other children, that they do go back a little bit before the summer holidays so that they can do some of those last memory bits that they wanted to do, the signing of the shirts and a bowling trip and the fun things that you do to mark finishing primary school." (P18)*

The limited entertainment options that families could do outside the home increased the activities they did within the home ('in-house activities'), such as arts and crafts, baking and reading. However, parents found it challenging to suggest new and interesting in-house activities; children had started to lose interest in these activities that were once new and exciting.

> *"we've probably done more activities in the last three weeks than probably what we've done in the past year, spread out over time."*

Parents reported that this was why some children struggled to get out of bed in the morning and suggested that there was *"nothing really that fun to do."* Parents reported, *"it's the same thing every single day,"* and *"there's only so much of that [playing in the garden], it's going to wear off after a bit."* Parents reported limiting their children's time on activities to prevent activities from being overused and children becoming bored with them. In addition, parents bought new toys and other items to keep children entertained throughout the day.

> *"I've found myself buying toys and . . . loads of arts and crafts. So, shopping trips are more expensive, but it's not the food that's making it more expensive." (P13)*

Parents seemed to reflect on items they had previously refused to buy for their children and decided to buy them to increase the number of in-house activities available. Parents were less concerned about buying activities solely for entertainment when the activity also incorporated a "new skill" that their child could learn. Parents commonly reported their children were playing with toys and games that had been left unopened for years, highlighting the amount children were playing within the home.

The lack of social interactions with friends and relatives was consistently reported as "very difficult," "most challenging," and "hard" for both parents and children.

*"He's OK 90% of the time, but he will have his moments where he . . . like the other day he woke up and the first thing he said to me was, 'I miss my friends' and the whole day he was just . . . sad, I think is probably the best way to describe it." (P12)*

Some parents suggested they were "overcompensating" with "treats" when their children were upset about the activities and social interactions that they were missing. Children were worried about the impacts of not being able to socialise with their friends, such as how friendship groups will "alter in dynamics," and upset that their "friends aren't talking" to them as much as other people in the same friendship group. Families commonly reported that they would regularly interact with their relatives because they would help with childcare. The change in routine made the lack of interactions more noticeable and difficult.

*"so that's been quite a major difference in my wife's weekly routine, and the girls in particular are absolutely missing their nan and granddad because they see them at least once a fortnight." (P25)*

Parents encouraged children's use of computers, mobile phones, and other electrical equipment that they could use to socialise with their friends. However, parents reported that using technology as a means of social support was less beneficial for children who needed supervision and were less fluent and relied on "play" to socialise.

*"She's had some Skypes with a few people, and some more than once, but it's really hard, she'll do the first couple or three minutes and then she's not really that interested any more. She's spoken to them, she's seen them, done! But if they were here, she'd be playing with them and interacting with them a lot more and in a lot different way." (P24)*

Some families socialised in-person, which helped keep children entertained and cope with friends and relatives that they missed. However, in-person socialising was irregular and mainly consisted of families having conversations at a two-meter distance, through windows and for short periods. Still, parents reported the benefits of these sporadic in-person interactions.

*"Obviously we have a conversation, but they stay just in the doorway, or through the window and we stay out on the pavement. One time we did that, we did take the kids because then we had our exercise in the playing field near them, and it's just nice for my dad and his wife to see the kids. We combined it with our exercise for that day, so that was quite nice." (P27)*

Three other factors appeared to help families, particularly children, cope with the lack of social interactions and entertainment activities. First, children interested in in-house activities before the lockdown were able to continue with their usual activities. Second, parents would encourage activities that the whole household could engage in, keeping children's interest in the activity for an extended time, and for everyone to enjoy the activity.

*"She just dances on her own . . . she does, what's it called, TikToks. We try to do that as a family. <Laughs> Me and her try to do it together. Sort of try and keep her a bit entertained." (P01)*

Third, after school clubs and activities such as sports and dance clubs, army and police cadets were cancelled. However, some of these activities moved online, which parents commonly found beneficial, and suggested children could engage with their school friends, stay entertained, and appeared less bored.

*"They all, actually the other thing that, to a lesser extent, but they're all members of sports clubs. The boys, both football; my eldest, rugby. T\*\*\* is football and hockey. And the coaches there . . . it's all voluntary, so the coaches have been pretty good at sending out video clips of some football skills, or whatever that might be. And again, challenges in a way, setting the kids the challenge of practicing those skills. . . And then all three of them will do that and do that in the garden. And then, we can video the skills and post it back on the WhatsApp group for the sports team." (P27)*

However, some activities could not be transferred online, some activities that were supposed to move online had not been organised, and some children found online activities less enjoyable. Similarly, parents tried to attend activities that had moved online but found it difficult to concentrate on the activity because they still had to care for their children.

*"I normally do a restorative yoga class on a Sunday night and I do miss going to that, to be honest. But we've got some classes online and I've been trying to keep up with that if I can. Again, it's trying to do a bit of yoga and you've got, otherwise I've got S\*\*\* running in saying he wants something, the dog's trying to knock me over, so it's not the most relaxing time." (P16)*

## 2.2 Physical activity

Most parents reported the importance of exercise and encouraged their children to exercise for their physical health and help them cope with lockdown guidance. Parents suggested that their children's mood and behaviour were adversely impacted when they did not exercise. In connection, parents suggested they wanted their children to exercise to "burn off energy" because it helped them maintain good behaviours.

*"Yeah, to be honest, he has been really good most of the time. Yesterday he'd kind of got one on him, I think it was difficult Saturday with the weather, with it raining because he couldn't go outside, so that kind of altered his mood as well. Because if he's in the garden, it's just trying to get all the energy off him as well, because normally we do a lot of activities as well, and obviously we can't do any of those at the moment and they tend to tire him out and obviously having too much energy affects his behaviour as well." (P16)*

Parents' reports varied about the amount of exercise or time their children spent outside compared to when schools were open. Some parents suggested their children were outside and exercising more such as spending most of the day in the garden, but some reported they were outside less. These parents were concerned about the impacts of their children being less active.

*"Yeah, yeah, so obviously they had PE through school which I think they do once a week, and then they've got an after-school sports as well, so they had two sports activities connected to school, which have ceased. They both, one attended football, yeah, he went to football training*

*twice a week, and then they both attended cross fit, which is, yeah, just like physical exercise. And then they both attended swimming on a weekly basis. So, yeah, one, two, three. So, if you're reducing it into time, it would be like four and a half hours of physical activity per week which is scheduled in effectively, that's not there anymore." (P17)*

The importance of a garden was vital for keeping their children active, and parents reported: *"it is a Godsend,"* and they were *"really fortunate"* and *"lucky"* to have a garden. When families did not have a garden, other outdoor areas were vital such as parks and families without access to outdoor spaces reported it was challenging to cope with staying inside. However, parents struggled to get their children to exercise regularly.

*"And then again just getting out the house, so OK, we need to go to Tesco Express and get some bread or milk, we'll do a long walk or bike ride round, just to get out of the house and back in and get that exercise, otherwise they would just literally stay in the house all day. . ." (P04)*

Parents would actively seek activities that provided exercise and that were similar to activities they did before the lockdown to try to motivate their children to keep active.

*"we often turn our lounge into basically a soft-play so we have a massive sofa, a big corner sofa, and we take the cushions off and put them all over the floor. . . I mean when we do it, he plays it on and off but when he really plays with it he must be 40 minutes, he's jumping over the cushions, jumping over. . ." (P07)*

Similarly, some parents would seek different walking routes and new games to play outside to try to keep their children's interest in activities that kept them active.

*"I'm renovating our garden so they'll come and get outside and get active and get muddy and stuff with me. They love being outdoors so they love getting engaged in the garden with plants. They've got a little climbing house which they play in and just like they do indoors with role-play and coming up with games and challenges with each other, they'll just continue that outside as well, which is really nice." (P25)*

Another method parents used to persuade their children to exercise and to get them out of the house was by emphasising other things about the activity that they may find interesting rather than focusing on exercise such as walking to visit friends and family, food shopping and walking the dog (within the guidance).

Furthermore, online resources helped parents keep their children active, including physical clubs and activities that had moved online. Exercising with "Joe Wicks" was common throughout families. Joe Wicks was a fitness coach who pledged to provide free online fitness classes throughout lockdown every day of the school week. However, the technology was not always available and practical for children to exercise using online resources.

*"we try to do the Joe Wicks ones, but I'm having to use my laptop for work and the spare laptop we had was awful at just, we couldn't see it. But yesterday I got my phone and I said, 'Right, run round laps and see how many steps you can get counted up on my phone' just to try and get him doing things. And we're quite lucky, we've got a bouncy castle, a little mini one, which we put out in the garden, so get him to run and jump on that and things." (P12)*

**Table 4. The factors that affected how families coped with the COVID-19 school closures and the lockdown guidance in relation to Theme 3: Worries about the COVID-19 pandemic.**

| Theme 3: Worries about the COVID-19 pandemic |
| --- |

**Sub-theme 3.1: Media and information**
• Families felt "anxious" and "depressed" when they viewed information about COVID-19, and therefore, parents tried to limit the amount of information they viewed about COVID-19.
• The lockdown felt "sudden" and the lack of key information (e.g., resources to teach their children, occupations considered critical to the response and who were eligible for furlough) that parents received increased their worries about lockdown.
**Sub-theme 3.2: Worries about health**
• Parents and children were burdened by the worry about their friends and family's health.
• Those perceived as "healthy" mitigated people's worries about their health, but this coping strategy was less effective for individuals with health conditions that made them vulnerable to COVID-19. The uncertainty about how severe symptoms of COVID-19 could be if they were infected was still a common worry.
• Maintaining their partners and their children's well-being was important to parents. Well-being was commonly considered a priority compared to home-schooling and other factors that may adversely affect their family's mental health.
• Families had mantras, which they used to help them cope with the lockdown
• Worries about standard health care not being available exacerbated parents' health worries.

Parents reported that they struggled to motivate themselves to exercise, even though they knew the benefits. Therefore, they would try to make an extra effort to encourage their family to exercise and not stay indoors.

**Theme 3: Worries about the COVID-19 pandemic.**    A summary of the factors described in theme 3 is present in Table 4.

### 3.1 Media and information

Parents were worried about the information their children received about COVID-19. Parents found the balance between informing their children about COVID-19 to mitigate their worries and keep them safe, whilst the information did not "scare" or "worry" them.

> *"My four-year-old doesn't understand anything about it. And like I said, with my eldest one, I have tried to shield her away from the news and stuff, 'cause it's not nice, but then again we have had to tell her about social distance and she needs to know about that, so out of ten understanding, she probably understands around six, if that makes any sense." (P10)*

Furthermore, parents were concerned about the information that their children had viewed in the media and learned from their teachers and friends. Some parents also reported that they had increased the amount they viewed media compared to before the closures to learn about COVID-19, which made them, and their children anxious and was "depressing" to watch. Parents felt overwhelmed with the amount of content about COVID-19 and thought it was "constant." Parents tried to limit how much time they and their family engaged with media about COVID-19 (e.g., only viewed information about COVID-19 at specific times or topics).

The Prime Ministers' announcement about the school closures and the lockdown guidance was "shocking" and "sudden." Parents commonly believed the Government were handling a complex and unprecedented situation, but they found the first couple of weeks the hardest due to a lack of guidance and support, particularly about home-schooling, financial aid for businesses and furlough.

> *"I kind of thought, that it was just all a bit rushed. I thought, 'Oh my God, schools are closing, there's not really been . . .' Obviously you can't really say anything, but we did kind of know*

*about coronavirus for a long time before they started doing anything about it. So, it was a bit bizarre. It was like all of a sudden, the schools . . . you can take your child out of school, or we're gonna close the schools anyway. But then some people can come in and some people can't come in. And it was just all really bizarre." (P13)*

Furthermore, the lack of information about how long the COVID-19 guidance would be in place was a constant source of stress *"I don't know! I'm gutted. I honestly don't know how long it's gonna last."*

## 3.2 Worries about health

Parents and children were worried about loved ones, particularly grandparents, because of their vulnerability to COVID-19, and they were upset about not being able to socialise with their grandchildren.

*"I was in bed the other day and he [child] said, 'Coronavirus is killing people, isn't it?' And I said, 'It is, yeah, but they tend to be very old or very poorly people and we're doing the best we can to stay safe and that's why we've got to stop in'. And he said, 'It's going to kill my nan and grandad, isn't it?'.. and that's really upsetting to deal with. . ." (P16)*

Parents were also worried about spreading COVID-19 to their loved ones.

*"he [partner] is like worried about catching coronavirus or passing it on to the kids or me, or me getting it and bringing it back in." (P04)*

In addition, parents were worried about becoming ill and being unable to adequately care for their children, although this was commonly reported as a concern in single parents.

*"that is one thing we have talked about because, deep down, that's a real worry for me because if I was ill, it's just me and the children, that is a real worry." (P29)*

Parents who perceived they or their family were "healthy," "rarely ill," and not "sickly," appeared to alleviate parents' worries about COVID-19 and equated good health with being *"hopeful that none of us will get it [COVID-19] quite bad."* Parents had a common perception that COVID-19 *"doesn't affect children as severely as it does adults,"* which they supported by "research" and "science" gained via the media. This perception also alleviated parents' worries about their children becoming severely ill with COVID-19. However, these mitigating factors were less effective in reducing parents' worry about their children's health when they had health conditions. Furthermore, parents reported that even though they perceived their children to be healthy, they worried about the uncertainty and suggested that their children could react badly and become unwell. Parents commonly supported this worry with the information they had viewed connected to young children who had died or become seriously ill with COVID-19.

*"I don't know . . . I mean it all worked very well until the first thirteen-year-old died and then it was just like 'Hold on a minute'. I thought that it didn't affect young people, only people with illnesses. . ." (P01)*

Parents were also concerned about their families' mental health, and suggested their priority was to ensure they were all *"happy, healthy and having the positive well-being,"* rather than

worry about home-schooling or other factors that may adversely impact them or their family's mental health.

> *"If they work hard, then they can. But at the moment they just need to relax, because relaxing will make them feel happy and it will be easier for them to get over the shock and do well at school, and be academically well, high achievers." (P22)*

Families commonly had a mantra that helped them cope with worries about COVID-19 and the struggle with staying at home such as: trying not to worry because there is nothing that parents can do to change the situation; focusing on religion (e.g., *"leave every other thing to God"*); believe there were worse situations that they could be in; *"we're all in this together;"* trying to find humour in the situation; *"accept it and get on with it;"* focusing on the positives and the privileges or fortunes the family have; and focusing on the future.

Parents' healthcare experience during the lockdown impacted the family's concerns about health. Some parents found that appointments with their GP were more efficient. However, some parents had vital appointments cancelled or delayed, which added to parents' health worries.

> *"we were so excited because it's taken 1.5 years to come through, because obviously the system takes that long because it's part of the child development, and then I got a phone call just before, as soon as this lockdown started. 'Sorry, his appointment has been cancelled because of the coronavirus. . .' And I said to them, 'When do we expect it?' They said it could take up to six months, it could take even longer than that." (P14)*

Parents who did not have first-hand experience with the healthcare system were commonly concerned about standard healthcare being unavailable and that they could not visit their child in hospital if they became unwell. Parents perceived the healthcare system as overwhelmed, their local hospitals had reached capacity and closed, and they were worried about seeking medical attention and exposing their family to COVID-19. Parents tried to prevent their children from hurting themselves to reduce their anxiety about seeking medical attention and burdening the healthcare system and commonly reported that they had told their children to be extra careful when they played. However, parents would seek medical attention if they needed to because they had "no choice."

## Discussion

In this work, we identified three themes and eight sub-themes that affected how families coped during England's school closures and first COVID-19 lockdown in relation to family well-being, children's physical activity and education. It has been suggested that the increase in mental health issues observed in families following a disaster is due to families not adapting to the change in circumstances because of the disaster [8]. Our findings build on this theory and identify the changes families experienced during the lockdown and factors that prevented and supported how family coping and adaption to the changes that the COVID-19 pandemic created. As such, some families had increased challenges due to pre-existing social and economic inequalities and factors outside of their control.

### Family well-being

Research suggests that school closures and lockdown guidance implemented during the COVID-19 pandemic was likely to adversely impact a family's well-being [7–9]. The

uncertainty and added stress of the pandemic risked creating hostile home environments, although families with high resilience might have been better able to cope with these challenges [8]. For the most part our findings support this. As expected, home-schooling was a primary cause of stress within the home followed by worries about family members and friends and managing the lack of social interactions and activities. Financial worries were also a main concern although only for some families; other families reported they were in a financially better position than before the pandemic. Parents also reported that living in close proximity with each other and without any respite also exacerbated what might have been minor arguments in other circumstances. However, parents also reported some positives about the school closures, including spending time with family, and reduced school pressures that helped to maintain a calming home environment, which was less considered in previous research [7–9].

However, a common cause of conflicts within the home was home-schooling disagreements, which stemmed from parents trying to teach their children. Parents commonly reported they did not have the skills to teach their children. This included parents who were qualified teachers but were not teaching in school due to the closures (e.g., on furlough or working reduced hours). Furthermore, we found that working parents reported having less time to support their children with home-schooling and their other needs, such as emotional support and development, which can adversely impact children's well-being [40]. Therefore, children who were engaged with home-schooling and were less dependent on parent and teacher instruction were more likely to be able to adapt to home-schooling, which will protect a family's well-being.

Families commonly struggled with COVID-19 guidance that prevented individuals from different households being able to meet in-person, which is in line with previous research finding associations between low social support and children's mental health problems [9]. Technology was used to mitigate the lack of social interactions but did not compensate for socialising in-person. A UK study conducted in the first 100 days of lockdown found that parents with school-aged children were at higher risk of loneliness [41]. Loneliness has also been found to increase risk for adverse health impacts, such as mortality in adults [42] and depression and anxiety in children [43,44]. After this study, in early June 2020, the Government relaxed the COVID-19 rules to support people living alone, which included single parents who were at risk of feeling "lonely and struggling" [45]. However, there were no such rules to mitigate loneliness in children, which is of concern given that the adverse mental health effects due to loneliness can impact children for years [44], this should be considered in future.

Parents commonly reported that they and their family were worried and upset about the health impacts that COVID-19 could have on loved ones. In connection, parental worries about their family's health were also a common reason why children eligible to attend school were home-schooled. An American study found that children increased telephone contact with their grandparents during the COVID-19 pandemic, and the most common reported reason was out of worry for their grandparents in the context of COVID-19 (74%) [46]. We suggest that health concerns about COVID-19 are exacerbated in families due to the elderly, who are often grandparents, being at risk for serious illness with COVID-19. In addition, parents' worries about ill health appeared to alleviate when they perceived their family as "healthy." It has been found that those who were employed during COVID-19 and had mental or physical health disabilities tended to have more concerns about health and finances, and perceived less organisational support compared to individuals without physical or mental health problems [47]. Parents also reported feeling anxious and depressed by the media coverage about COVID-19. The term "doomscrolllling" has become a common term used to describe individuals who spend an excessive amount of screen time on negative news, and has been shown to

increase the risk of depression and post-traumatic stress disorder [48]. Therefore, we suggest that families who limited their screen time about COVID-19 may have facilitated their ability to cope with the lockdown and children and parents should be informed of the potential adverse consequences of too much screentime.

It is common for research to centre on the impact of parents' well-being on the family system [8,9]. We found that parents reported that their children's mood impacted the mood of the household and it was common for parents to report that their children's mood had improved due to reduced school pressures, which they commonly reported resulted in a better home environment than before the lockdown. Parents also reported some positive behaviour changes in their children. Families being able to spend time together that led to them understanding each other better was an apparent facilitator to coping with the lockdown, supporting research about the importance of the parent-child relationship in determining well-being [49,50]. In addition, feeling connected with caregivers has been found to predict child happiness [51], with good communication between parents and children being a protective factor for child mental health [52]. However, parent's mood was also important. Parents who felt they were unable to enjoy the interactions with their children during lockdown were more stressed and less able to cope than other parents, which supports current research [53]. And mirrors our findings about working parents who felt burdened by not being able to spend time with their children because they were working.

Furthermore, the pressure that parents placed on themselves to ensure that they were doing right by their children, exemplified in the theme of "parent guilt," was common throughout our findings. Research links feelings of guilt with depression [54,55]. In addition, Wright [56] analysed the work of Maushart (1999), *The Mask of Motherhood*, which describes a common ideology about mothers and the perception that to be a "good" mother, you cannot show negative emotions, such as anger, and instead mothers hide behind a "mask of motherhood" to try to display to others the "perfect" mother rather than their parenting struggles. It could be suggested that this ideology may be outdated, particularly with the emergence of websites where parents openly discuss the *imperfect* realities of being a parent [57,58]. However, we found that parents were trying to hide their emotions and were "putting on a brave face" in front of their children and via school platforms. We suggest that feelings of guilt and masking negative emotions, if experienced for an extended period, would have had an impact on the family system, in line with previous research about parental stress filtering into other household members [8].

But, overall, parents commonly reported that their family was coping well with the lockdown. This finding mirrors another study that found that 71% of UK parents coped with the lockdown [59], indicating that for most, the mental health risks may be less severe than expected. However, most of the families in the study also felt that they had only been mildly impacted by the pandemic. Support might be best targeted to families who are most at risk, such as those who had experienced grief, severe financial consequences and had pre-existing health conditions that made them vulnerable to COVID-19, or to children who were already at risk, such as from poverty, neglect and abuse [60].

## Physical activity

Children are recommended to engage in 60 minutes of physical activity a day to enhance cardio-metabolic health, mental well-being and improve their confidence and peer acceptance [12]. We found that parents struggled to motivate themselves and their children to exercise, particularly when they had limited access to outside space and were worried about COVID-19. However, parents commonly tried to encourage their children to exercise and believed

exercise helped to maintain good mental health. We suggest that this perception has protecting benefits, for both parents and children's physical and mental health. Research has shown that children and adults who engaged in physical activity during COVID-19 had improved well-being [61,62].

Families with access to a garden and outdoor space were more active. We observed a difference between the amount of exercise within families. Some children spent most of their time outside and were active while other families engaged in physical activity for less than an hour a day. This tallies with work by others, which showed that 21% of children exercised more than usual, 27% the same, and 52% less than usual during lockdown [63]. That report also cited that a "lack of access to their usual space or place" and "concerns about the virus" were key barriers preventing children from exercising, which supports our findings. Again, there is an element of health inequality apparent in these findings, with lack of access to outdoor space during lockdown potentially exacerbating the existing links between childhood obesity and poverty [64], and mental health [65].

### Education

Children and parent characteristics played an important part in how well the family could adapt to home-schooling. Our findings support the concerns that were raised in opposition to the school closures [16,25–27]. On average, children received about 2–3 hours education at home, less than was recommended by Government [29], although these findings need to be taken with caution. Parents struggled to report the amount of time they spent teaching and suggested they incorporated education into most activities even when they were not "home-schooling." Parents with children in primary school had to support and supervise their children more than parents with older children. This finding supports a study in England about home-schooling, which found that 60% of learning at Key Stage 1 (five to seven-year-olds) was dependent on parental instruction compared to 30% for Key Stage 2 (seven to eleven-year-olds) [66]. This suggests that families with young children without parental support may be more at risk of being behind in their education, compared to older children, such as teenagers or children with parental support. However, mothers and fathers took on an additional three and a half hours of childcare and educational responsibilities a day than before the pandemic [67]. But, regardless of work status, the time spent on these responsibilities disproportionately impacted mothers because they were already spending more time on housework and childcare than fathers before the pandemic [67]. In addition, some children completed their schoolwork with minimal supervision and guidance from their parents or the school, whereas for other children, parents struggled to get their children to engage with schoolwork. Moreover, children (in primary and secondary school) from middle and poorest income households have been reported to have less support (e.g., online classes, school work, and private tutor) with their education compared to children from higher income households [68]. Motivation is a key element to learning and research suggests that optimum learning results from teachers being creative to keep children motivated [69]. Children may have benefitted from parents who opted for a child-led teaching approach to maintain their children's motivation, which may be particularly beneficial for children who do not engage as well in the traditional school system [70].

The Children's Commission was worried about the disadvantage gap widening between children due to the pandemic, and our findings suggest this is likely [71]. The importance of access to resources in how families were able to adapt to home-schooling and cope with the lockdown was a theme throughout the study, which supports previous research that associates low social economic status with poorer education [19–21]. Home-schooling was also reliant on stable internet connections, which will have larger impacts on families living in rural areas

[72]. In line with this, other studies have shown that families that were financially stable were better able to cope with lockdown compared to families that were less financially stable [73,74]. We also identified that children with special educational needs and/or disabilities (SEND) were at increased risk of social and educational disadvantages due to their specialist education ceasing during lockdown. A Government report released in May 2021 on schools before the pandemic showed that SEND children were already receiving a lack of education tailored to their specific needs [75]. The factors that we have identified, such as the impact of children's level of motivation, parents not having the skills to teach or being able support their children during home-schooling could explain the recent reports that suggest children are between two and three months behind in their education due to the school closures [76] and parental belief (63%) that their children's education suffered during the pandemic [59].

Aside from the challenges we identified with home-schooling, some parents appreciated the autonomy they had to teach their children what they wanted, which included educational topics and skills to develop them into well-rounded people. Many parents believe that some areas are not adequately taught in schools, with one recent survey finding that 63% of parents felt that schools badly prepared children for life in general, and 53% of parents reported that the national curriculum should change [59]. Home schooling allowed some parents the opportunity to correct these perceived deficiencies.

## Implications and limitations

Systems need to be in place to support parents in future school closures to mitigate the mental, physical, and educational impacts on children, parents, and the family system. In this respect, furlough was a key resource to provide financial security to parents who could not work and allow parents to care for their children. Similarly, parents who worked in supportive organisations were better able to adapt to lockdown. Organisations need to improve their emergency planning procedures to implement emergency measures more efficiently in the future, particularly for parents who have mental and physical health problems. We suggest that future research focuses on designing interventions that enhance the benefits of "family time" to minimise parent's guilt and stressors. A second problem was that parents had to educate their children without the skills or guidance to home-school. Schools that provided tailored support to parents and children were beneficial and, in turn, facilitated children's education and supported parents' and children's mental health. Third, parents appear to be ill-informed about their children's education.

This study had limitations. First, parents who opted to participate may have been particularly motivated to participate in a study about lockdown, which may limit our findings. These parents may have similar beliefs and motivations about the lockdown guidance and home-schooling. Second, most parents were married or cohabiting (70%) and white (67%). Therefore, further research is needed in relation to lone parent and ethnic minority households. Third, the interviews were conducted at the start of the school closures and lockdown measures. Things may have been different later on in the pandemic. However, it is important to see how families coped with the initial disruption and in a time of considerable uncertainty, to be able to better prepare for any future sudden school closures. Still, it would be beneficial to investigate the study's findings in a follow-up study to identify the longer-term impacts of the school closures and lockdown guidance.

## Conclusion

It is clear that schools being closed and the lockdown guidance impacted families, due to home-schooling and parents having to care for their children whilst they had other

responsibilities and burdens such as financial and work commitments. However, families appeared to adapt to the difficult situation. Having more resources, including equipment to home-school and adequate time and skills to facilitate home-schooling mitigated the mental health burden on families. In addition, children who could educate themselves and were not reliant on their parents for support were better able to cope with home-schooling. Parents and children who were connected to their family members and engaged in physical activity were better able to protect their well-being.

## Supporting information

**S1 Text. The discussion guide.**
(DOCX)

## Author Contributions

**Conceptualization:** Lisa Woodland, Rebecca K. Webster, Richard Amlôt, Louise E. Smith, G. James Rubin.

**Data curation:** Lisa Woodland.

**Formal analysis:** Lisa Woodland.

**Funding acquisition:** Rebecca K. Webster, Richard Amlôt, G. James Rubin.

**Investigation:** Lisa Woodland, Ava Hodson.

**Methodology:** Lisa Woodland, Richard Amlôt, Louise E. Smith, G. James Rubin.

**Project administration:** Lisa Woodland.

**Resources:** G. James Rubin.

**Supervision:** Rebecca K. Webster, Richard Amlôt, G. James Rubin.

**Visualization:** Lisa Woodland.

**Writing – original draft:** Lisa Woodland.

**Writing – review & editing:** Ava Hodson, Rebecca K. Webster, Richard Amlôt, Louise E. Smith, G. James Rubin.

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
