## [Decision Letter · Decision Letter 0]

29 Sep 2022

PONE-D-22-21321How families coped with the COVID-19 school closures in England: a qualitative study about parents’ battles with home-schooling and the factors that impacted children’s education, physical activity, and well-beingPLOS ONE

Dear Dr. Woodland,

Thank you for submitting your manuscript to PLOS ONE. After careful consideration, we feel that it has merit but does not fully meet PLOS ONE’s publication criteria as it currently stands. Therefore, we invite you to submit a revised version of the manuscript that addresses the points raised during the review process.

We look forward to receiving your revised manuscript.

Kind regards,

Jennifer Coto, PhD

Academic Editor

PLOS ONE

Journal Requirements:

   "This study was funded by the Economic and Social Research Council [grant number ES/P000703/1] and National Institute for Health and Care Research Health Protection Research Unit (NIHR HPRU) in Emergency Preparedness and Response, a partnership between the UK Health Security Agency, King’s College London and the University of East Anglia. The views expressed are those of the author(s) and not necessarily those of the 

NIHR, UKHSA or the Department of Health and Social Care. For the purpose of open access, the author has applied a Creative Commons Attribution (CC BY) licence to any Author Accepted Manuscript version arising."

   "This study was funded by the Economic and Social Research Council [grant number ES/P000703/1] and National Institute for Health and Care Research Health Protection Research Unit (NIHR HPRU) in Emergency Preparedness and Response, a partnership between the UK Health Security Agency, King’s College London and the University of East Anglia. The views expressed are those of the author(s) and not necessarily those of the NIHR, UKHSA or the Department of Health and Social Care. For the purpose of open access, the author has applied a Creative Commons Attribution (CC BY) licence to any Author Accepted Manuscript version arising.

The funders had no role in study design, data collection, data analysis, data interpretation, or writing of the manuscript. The corresponding author had full access to all the data and had final responsibility for the decision to submit for publication."

7. Please amend the manuscript submission data (via Edit Submission) to include author Louise E Smith and G James Rubin.

Additional Editor Comments:

Please address the reviewers comments to improve the suitability of your manuscript. Importantly, place emphasis on the introduction so that it is more consist with the aims and rationale for the study. 

Reviewers' comments:

Reviewer's Responses to Questions

**Comments to the Author**

1. Is the manuscript technically sound, and do the data support the conclusions?

Reviewer #1: Partly

Reviewer #2: Partly

2. Has the statistical analysis been performed appropriately and rigorously? 

Reviewer #1: N/A

Reviewer #2: Yes

3. Have the authors made all data underlying the findings in their manuscript fully available?

Reviewer #1: Yes

Reviewer #2: Yes

4. Is the manuscript presented in an intelligible fashion and written in standard English?

Reviewer #1: No

Reviewer #2: Yes

5. Review Comments to the Author

Reviewer #1: The title should be revisited. It is long-winded and not focused.

The background section in the abstract says that “there is limited understanding about the impacts of this (school closure in England) on children’s mental and physical health and their education.” But then the next sentence goes on to say that you therefore decided (logical to assume based on the preceding statement) to explore "how families coped during the school closure". One expects that the study should have explored the issues such as children's mental, physical health and education, as it seems to be the gap identified.

Further, while the background suggests that the study wanted to establish "how" families coped, in the results, the it says the “themes that impacted a family’s ability to cope during the lock-down. Based on the background, one expects to read on “HOW families coped”. However, the results suggest that attention is on “Factors that impacted a family’s ability to cope”. Furthermore, the conclusion also talks about the “factors that impeded families to cope with lock-down”.

It is highly recommended that you should decide on a particular issue, be it "how families cope", family's ability to cope" or "factors that impeded family's to cope". Another way, would be to revisit the main objective and therefore, the title of the study and have the three as MAIN themes.

Just to make the point clearer, also, the title suggests the study was about the “factors that impacted children’s education, physical activity and well-being”.

It is important that the paper should focus on a particular issue and ensure a “golden thread”. The current version seems to wonder from one issue to another. It also makes conclusions not based on the results presented.

Reviewer #2: Manuscript Review for PLOS ONE (Manuscript ID: PONE-D-22-21321)

The overall aim of the manuscript entitled “How families coped with the COVID-19 school closures in England: a qualitative study about parents’ battles with home-schooling and the factors that impacted children’s education, physical activity, and well-being” was to examine factors that impacted i) a family's well-being, ii) children’s education, and iii) physical activity during the school closures in England, and how these factors affected the ability of families to cope with the COVID-19 pandemic. Overall, the manuscript has notable strengths, however, there are several concerns that limit the enthusiasm about the current manuscript and should be addressed.

Overall:

1. Please review the manuscript to ensure it follows APA formatting.

2. The manuscript uses the terms “parent” and “caregivers.” This reviewer would encourage the authors to use the more inclusive term (caregivers) throughout.

3. Generally, the manuscript could also benefit from stronger topic sentences and transitions throughout that better reflect the information covered within the paragraph and enhance the fluidity of the manuscript and clearly illustrate the information for the readers as they are going through the manuscript. For example, the second paragraph in the introduction discusses home-schooling and how families adapted. However, the third paragraph discusses the benefits of schooling, such as “provid[ing] children with access to health services…” A stronger transition between the two paragraphs is recommend.

Introduction:

4. Overall, the introduction would benefit from being expanded and edited to better set the stage for the current study. For example, the manuscript would benefit from a brief review of the potential long-term mental health implications for children and families. Further, the impact of other familial and environmental factors (e.g., SES, race/ethnicity, language, education level, preexisting emotional and behavioral problems and/or caregivers with preexisting psychological concerns, etc.) is not discussed in the introduction. This is especially important given that minoritized and under resourced communities have been shown to be disproportionately impacted by COVID-19. This is alluded to on page 5, where the manuscript states: “children in low-income households can access free school meals,” but should be discussed in more detail as it is an important factor in understanding how families coped with the COVID-19 school closures. It would be helpful to expand the introduction to include a full review of the literature.

5. Additionally, it would be helpful to elaborate on the paragraph on page 5, “The impacts of the lockdown measures, on the family system, physical and mental health, and education, are not yet fully understood.” Specifically, there is literature indicating that children exposed to traumatic and/or stressful events, such as COVID-19, are at greater risk for experiencing emotional and behavioral problems (this is briefly mentioned in the discussion). Similarly, COVID-19 pandemic has had a negative impact on the level of stress and mental health of caregivers. In its current form, the manuscript does not outline a clear study scope or provide a rationale for why the study is needed and how it contributes and/or extends the current literature.  

6. The introduction would benefit from editing the current study section (page 5) to include a clear rationale for the current study along with outlined hypotheses.

7. The manuscript would benefit from defining on the following terms such as: “health care plan,” “Furlough,”” family's well-being,”” children’s education,” and “physical activity.”

Materials and Methods:

8. Additional information is needed regarding how “one-to-one qualitative interviews” were described to participants.

9. Eligibility criteria was described as “Participants were over 18 years of age, lived in England, and were the primary caregiver to at least one child (18 years and under) who was not attending childcare, pre-school, or school due to COVID-19.” Could they have other children attending school and if so, when responding to questions were they asked to consider the child who does not attend school? Also, were participants were receiving other services? These would be important to know to put the current paper in context, as some of these services may inherently impact study outcome variables. 

10. A weakness of the manuscript is the limited description of the recruitment (e.g., consent process, context, how the study was described to potential participants) and assessment process. This section would benefit from including a timeline of assessments relative to the consent process and study’s duration. Additional information when (relative to study timeline) did participants dropout  should be noted.

11. On page 6 the manuscript states, “Five hundred and thirty-nine potential participants applied to participate in the study and were screened for eligibility, and 47 were selected for follow-up screening via telephone.” Detailed information was not provided regarding participants that did not meet eligibility, agree to participate, or were excluded from the study. Were these participants different from those that continued their participation based on demographic information? It would also be helpful to know whether participants (if any) provided any reasons for not participating in the study. 

12. Similarly, on page 6 the manuscript states: “Eligible participants were selected for interview according to gender, ethnicity, marital status, employment status, income, level of education, living region, keyworker status, the number of children in the household, and children’s age to ensure a diverse sample." Additional information is needed to understand this additional screening process. It seems like 47 were selected for follow-up screening via telephone and only 30 were interviewed. It is not clear what screen criteria were for the different screening phases.

13. A rationale should be included for selection of the study age range given potential age-related differences in outcomes.

14. On page 6 the manuscript states, “We used a semi-structured interview guide to explore how families coped with the school closures.” It would be helpful to provide additional information about this approach- was the interview developed? If so, by who? Was it based on an existing measure? How were questions created?

15. Similarly, the manuscript states “Four parents who had children in school or childcare before the school closures reviewed our initial interview guide. We amended questions, clarifying those that appeared challenging to answer based on their feedback.” The manuscript would benefit from providing information related to how those four families were selected. What were the ages of their children? This is important given potential age-related differences in how families experienced and coped with the COVID-19 school closures. This should also be noted as a limitation.

16. The manuscript states, “All interviews were audio-recorded and transcribed verbatim.” It is important to describe how this process was explained to participants to ensure participants felts comfortable disclosing more detailed and in-depth information. Also, who transcribed the interviews?

17. It is unclear whether all authors were involved in creating the initial topic groups? Were any other study team members involved in this process?

Results:

18. On page 8 the manuscript states, “Further demographic information is presented in Table 3. All participants had at least one child who was not attending school or childcare because of the pandemic, although six were not in childcare or school before the closures.” It would be helpful to include this point in the limitations section given that families of children who were not in childcare or school before the closures may differ in terms of existing resources and potentially experiences and coping strategies.

19. It would be helpful to add the percentages to the demographic table on page 8.

Discussion:

20. The authors should consider revising this section for improved clarity overall. It reads in some areas like an Introduction section. It would be helpful for the discussion to include a more in-depth discussion about the generalizability of these findings and interpretation of the findings that includes possible explanations and implications, as well as how the current study extends the work has been done by clearly outlining clinical and research implications of these findings. Lastly, how hypotheses were consistent/inconsistent with findings.

21. Limitations and future directions were brief. Consider expanding upon the limitations (e.g., majority of children were under the age of 12, lack of other caregiver involvement or report, limited generalizability) and future directions.

22. The manuscript notes, “Our findings highlight that some families had increased challenges due to pre-existing social and economic inequalities and factors outside of their control.” Discussion around pre-existing social and economic inequalities should be added in the introduction section.

6. PLOS authors have the option to publish the peer review history of their article (what does this mean?). If published, this will include your full peer review and any attached files.

Reviewer #1: **Yes: **Eric Umar

Reviewer #2: No

---

## [Author Response · Author response to Decision Letter 0]

5 Nov 2022

Comments to the Author

1. Is the manuscript technically sound, and do the data support the conclusions?

Reviewer #1: Partly

Reviewer #2: Partly

2. Has the statistical analysis been performed appropriately and rigorously? 

Reviewer #1: N/A

Reviewer #2: Yes

3. Have the authors made all data underlying the findings in their manuscript fully available?

Reviewer #1: Yes

Reviewer #2: Yes

4. Is the manuscript presented in an intelligible fashion and written in standard English?

Reviewer #1: No

Reviewer #2: Yes

5. Review Comments to the Author

Reviewer #1: The title should be revisited. It is long-winded and not focused.

The background section in the abstract says that “there is limited understanding about the impacts of this (school closure in England) on children’s mental and physical health and their education.” But then the next sentence goes on to say that you therefore decided (logical to assume based on the preceding statement) to explore "how families coped during the school closure". One expects that the study should have explored the issues such as children's mental, physical health and education, as it seems to be the gap identified.

Further, while the background suggests that the study wanted to establish "how" families coped, in the results, the it says the “themes that impacted a family’s ability to cope during the lock-down. Based on the background, one expects to read on “HOW families coped”. However, the results suggest that attention is on “Factors that impacted a family’s ability to cope”. Furthermore, the conclusion also talks about the “factors that impeded families to cope with lock-down”.

It is highly recommended that you should decide on a particular issue, be it "how families cope", family's ability to cope" or "factors that impeded family's to cope". Another way, would be to revisit the main objective and therefore, the title of the study and have the three as MAIN themes.

Just to make the point clearer, also, the title suggests the study was about the “factors that impacted children’s education, physical activity and well-being”.

It is important that the paper should focus on a particular issue and ensure a “golden thread”. The current version seems to wonder from one issue to another. It also makes conclusions not based on the results presented.

Response: Thank you for your comments, we have amended the title to “A qualitative study about how families coped with managing their well-being, children’s physical activity and education during the COVID-19 school closures in England.” This title is more concise and follows the aims of our study. We have also clarified throughout the paper that our aims of the study are to explore “how families coped” with the three main topics that we are investigating. 

In addition, we value the feedback that you have provided and as a result we have amended the structure of the paper (aims, results and discussion) to ensure that the manuscript reflects our study aims and that they thread throughout the paper. We have acknowledged this in the methods section, “feedback from peer review resulted in a change in how the themes were structured.” We have also added sub-headings to the discussion.

Reviewer #2: Manuscript Review for PLOS ONE (Manuscript ID: PONE-D-22-21321)

The overall aim of the manuscript entitled “How families coped with the COVID-19 school closures in England: a qualitative study about parents’ battles with home-schooling and the factors that impacted children’s education, physical activity, and well-being” was to examine factors that impacted i) a family's well-being, ii) children’s education, and iii) physical activity during the school closures in England, and how these factors affected the ability of families to cope with the COVID-19 pandemic. Overall, the manuscript has notable strengths, however, there are several concerns that limit the enthusiasm about the current manuscript and should be addressed.

Overall:

1. Please review the manuscript to ensure it follows APA formatting. 

Response: We are following PLOS ONE guidelines that does not allow APA formatting. 

2. The manuscript uses the terms “parent” and “caregivers.” This reviewer would encourage the authors to use the more inclusive term (caregivers) throughout.

Response: We have amended our manuscript to state that participants were eligible for the study if they “had parental responsibility for at least one child,” and removed the term caregiver. We need to identify that we interviewed people who had the right to make medical and educational decisions for a child i.e., they had parental responsibility. A caregiver may not necessarily have these rights. We acknowledge that “parent” is not completely accurate either, but it is most appropriate for a study and therefore we have kept this term throughout the manuscript.

3. Generally, the manuscript could also benefit from stronger topic sentences and transitions throughout that better reflect the information covered within the paragraph and enhance the fluidity of the manuscript and clearly illustrate the information for the readers as they are going through the manuscript. For example, the second paragraph in the introduction discusses home-schooling and how families adapted. However, the third paragraph discusses the benefits of schooling, such as “provid[ing] children with access to health services…” A stronger transition between the two paragraphs is recommend.

Response: We have tried to improve the transition between paragraphs whilst keeping concise throughout the manuscript. 

Furthermore, a similar comment was raised by reviewer 1 and in response we have amended our study aims and results to ensure that the manuscript follows an “golden thread.” We have acknowledged this in the methods section, “feedback from peer review resulted in a change in how the themes were structured.” We have also added sub-headings to the discussion.

Introduction:

4. Overall, the introduction would benefit from being expanded and edited to better set the stage for the current study. For example, the manuscript would benefit from a brief review of the potential long-term mental health implications for children and families. Further, the impact of other familial and environmental factors (e.g., SES, race/ethnicity, language, education level, preexisting emotional and behavioral problems and/or caregivers with preexisting psychological concerns, etc.) is not discussed in the introduction. This is especially important given that minoritized and under resourced communities have been shown to be disproportionately impacted by COVID-19. This is alluded to on page 5, where the manuscript states: “children in low-income households can access free school meals,” but should be discussed in more detail as it is an important factor in understanding how families coped with the COVID-19 school closures. It would be helpful to expand the introduction to include a full review of the literature.

Response: We have added more literature to the introduction to describe the potential long-term impacts of COVID-19 on families and the risk factors associated with the potential effects.

5. Additionally, it would be helpful to elaborate on the paragraph on page 5, “The impacts of the lockdown measures, on the family system, physical and mental health, and education, are not yet fully understood.” Specifically, there is literature indicating that children exposed to traumatic and/or stressful events, such as COVID-19, are at greater risk for experiencing emotional and behavioral problems (this is briefly mentioned in the discussion). Similarly, COVID-19 pandemic has had a negative impact on the level of stress and mental health of caregivers. In its current form, the manuscript does not outline a clear study scope or provide a rationale for why the study is needed and how it contributes and/or extends the current literature.  

Response: We have included more research to describe the impacts of stressful events on children and parents. 

6. The introduction would benefit from editing the current study section (page 5) to include a clear rationale for the current study along with outlined hypotheses.

Response: We have clarified the title, study aims and hypotheses. 

7. The manuscript would benefit from defining on the following terms such as: “health care plan,” “Furlough,”” family's well-being,”” children’s education,” and “physical activity.”

Response: We have defined health care plan and furlough in the introduction. Our definition of family well-being is deliberately broad to allow participants to express the beliefs that they feel will be informative. We do not feel that definitions of education and physical activity are necessary. 

Materials and Methods:

8. Additional information is needed regarding how “one-to-one qualitative interviews” were described to participants.

Response: Participants were informed that if they choose to take part, they would be invited to attend a telephone interview that would last no longer that 90 minutes and they would be asked a series of questions about how they have been managing since the school closures, which we have described in the procedure section. We have used the term “one-to-one qualitative interviews” in the manuscript to describe the study design method that was used.

9. Eligibility criteria was described as “Participants were over 18 years of age, lived in England, and were the primary caregiver to at least one child (18 years and under) who was not attending childcare, pre-school, or school due to COVID-19.” Could they have other children attending school and if so, when responding to questions were they asked to consider the child who does not attend school? Also, were participants were receiving other services? These would be important to know to put the current paper in context, as some of these services may inherently impact study outcome variables. 

Response: Our aims were to understand how the family as a whole were coping and they were not limited to talking about one child although the focus of the study was on children and their families who were not attending school. In your scenario we would have explored the reason for why some of their children were attending school but not others in case the reason related to our study aims, such as the parent could only manage supervising one child, so the other children went to school. The participants were not screened for services that they or their children were receiving although they were asked within the interviews about the physical, mental, and educational activities that they were engaging in. 

10. A weakness of the manuscript is the limited description of the recruitment (e.g., consent process, context, how the study was described to potential participants) and assessment process. This section would benefit from including a timeline of assessments relative to the consent process and study’s duration. Additional information when (relative to study timeline) did participants dropout should be noted.

Response: We have provided additional information about how long the interviews were advertised, interview process and dropouts. 

11. On page 6 the manuscript states, “Five hundred and thirty-nine potential participants applied to participate in the study and were screened for eligibility, and 47 were selected for follow-up screening via telephone.” Detailed information was not provided regarding participants that did not meet eligibility, agree to participate, or were excluded from the study. Were these participants different from those that continued their participation based on demographic information? It would also be helpful to know whether participants (if any) provided any reasons for not participating in the study. 

Response: We have clarified the recruitment process.

12. Similarly, on page 6 the manuscript states: “Eligible participants were selected for interview according to gender, ethnicity, marital status, employment status, income, level of education, living region, keyworker status, the number of children in the household, and children’s age to ensure a diverse sample." Additional information is needed to understand this additional screening process. It seems like 47 were selected for follow-up screening via telephone and only 30 were interviewed. It is not clear what screen criteria were for the different screening phases.

Response: We have clarified the recruitment process.

13. A rationale should be included for selection of the study age range given potential age-related differences in outcomes.

Responses: We have added to the introduction that the aim of the study was to understand the family experience, which included all school aged children. 

14. On page 6 the manuscript states, “We used a semi-structured interview guide to explore how families coped with the school closures.” It would be helpful to provide additional information about this approach- was the interview developed? If so, by who? Was it based on an existing measure? How were questions created?

Response: We have added detail about how the interview guide was designed. 

15. Similarly, the manuscript states “Four parents who had children in school or childcare before the school closures reviewed our initial interview guide. We amended questions, clarifying those that appeared challenging to answer based on their feedback.” The manuscript would benefit from providing information related to how those four families were selected. What were the ages of their children? This is important given potential age-related differences in how families experienced and coped with the COVID-19 school closures. This should also be noted as a limitation.

Response: We have included that the parents were recruited by the authors. The age-related differences are not considered a limitation - understanding these differences is interesting in its own right. 

16. The manuscript states, “All interviews were audio-recorded and transcribed verbatim.” It is important to describe how this process was explained to participants to ensure participants felts comfortable disclosing more detailed and in-depth information. Also, who transcribed the interviews?

Response: The participants were informed that the interviews were being recorded and shared with an external transcription company. Written and verbal consent was provided by all participants. Although we did not explicitly probe as to whether participants were comfortable with this, we use this procedure for many of our studies and have never had any incidents where participants appeared unduly concerned about this. 

17. It is unclear whether all authors were involved in creating the initial topic groups? Were any other study team members involved in this process?

Response: We have clarified how many authors were involved in the analysis. 

Results:

18. On page 8 the manuscript states, “Further demographic information is presented in Table 3. All participants had at least one child who was not attending school or childcare because of the pandemic, although six were not in childcare or school before the closures.” It would be helpful to include this point in the limitations section given that families of children who were not in childcare or school before the closures may differ in terms of existing resources and potentially experiences and coping strategies.

Response: We do not view this as a limitation but an aim of the study. We purposefully chose not to exclude these families or their responses from the study. All the participants that were included in the study had at least one child who was in childcare or school before the school closures and were not attending school because of the school closures. Six children were not in childcare or school before the closures (e.g., a newborn baby) but were included because their family met the eligibility criteria. We have made it clear to the reader that these children were included in the study and of the age of the children that were included. 

19. It would be helpful to add the percentages to the demographic table on page 8.

Response: We have added percentages to the demographic table. 

Discussion:

20. The authors should consider revising this section for improved clarity overall. It reads in some areas like an Introduction section. It would be helpful for the discussion to include a more in-depth discussion about the generalizability of these findings and interpretation of the findings that includes possible explanations and implications, as well as how the current study extends the work has been done by clearly outlining clinical and research implications of these findings. Lastly, how hypotheses were consistent/inconsistent with findings.

Response: We have updated the discussion. 

21. Limitations and future directions were brief. Consider expanding upon the limitations (e.g., majority of children were under the age of 12, lack of other caregiver involvement or report, limited generalizability) and future directions.

Response: We have added more detail to the limitations. 

22. The manuscript notes, “Our findings highlight that some families had increased challenges due to pre-existing social and economic inequalities and factors outside of their control.” Discussion around pre-existing social and economic inequalities should be added in the introduction section.

Response: We have added information about existing social and economic inequalities to the introduction.

---

## [Decision Letter · Decision Letter 1]

6 Dec 2022

A qualitative study about how families coped with managing their well-being, children’s physical activity and education during the COVID-19 school closures in England

PONE-D-22-21321R1

Dear Dr. Woodland,

We’re pleased to inform you that your manuscript has been judged scientifically suitable for publication and will be formally accepted for publication once it meets all outstanding technical requirements.

Kind regards,

Prabhat Mittal, Ph.D.

Academic Editor

PLOS ONE

Additional Editor Comments (optional):

The authors have been adequately addressed 

Reviewers' comments:

Reviewer's Responses to Questions

**Comments to the Author**

1. If the authors have adequately addressed your comments raised in a previous round of review and you feel that this manuscript is now acceptable for publication, you may indicate that here to bypass the “Comments to the Author” section, enter your conflict of interest statement in the “Confidential to Editor” section, and submit your "Accept" recommendation.

Reviewer #1: All comments have been addressed

2. Is the manuscript technically sound, and do the data support the conclusions?

Reviewer #1: Partly

3. Has the statistical analysis been performed appropriately and rigorously? 

Reviewer #1: N/A

4. Have the authors made all data underlying the findings in their manuscript fully available?

Reviewer #1: Yes

5. Is the manuscript presented in an intelligible fashion and written in standard English?

Reviewer #1: Yes

6. Review Comments to the Author

Reviewer #1: There is need to clean in text referencing. There are many instances where the reference has been put in a wrong place.

7. PLOS authors have the option to publish the peer review history of their article (what does this mean?). If published, this will include your full peer review and any attached files.

Reviewer #1: **Yes: **Eric Umar

---

## [Editor Report · Acceptance letter]

12 Dec 2022

PONE-D-22-21321R1 

A qualitative study about how families coped with managing their well-being, children’s physical activity and education during the COVID-19 school closures in England 

Dear Dr. Woodland:

I'm pleased to inform you that your manuscript has been deemed suitable for publication in PLOS ONE. Congratulations! Your manuscript is now with our production department. 

Kind regards, 

on behalf of

Dr. Prabhat Mittal 

Academic Editor

PLOS ONE